# Curvature Tuning: Provable Training-free Model Steering From a Single Parameter

**Leyang Hu**[*]
Brown University
leyang_hu@brown.edu

**Matteo Gamba**[*][†]
KTH
mgamba@kth.se

**Randall Balestriero**
Brown University
randall_balestriero@brown.edu

## Abstract

The scaling of model and data sizes has reshaped the AI landscape, establishing finetuning pretrained models as the standard paradigm for solving downstream tasks. However, dominant finetuning methods typically rely on weight adaptation, often lack interpretability, and depend on heuristically chosen hyperparameters. In this paper, we take a different perspective and shift the focus from weights to activation functions, viewing them through the lens of spline operators. We propose Curvature Tuning (CT), an interpretable and principled steering method that modulates a model's decision boundary by injecting a single hyperparameter into its activation functions. We show that CT provably adjusts model decision boundary curvature and, more fundamentally, projects a model onto a space of smooth functions—thereby complementing current finetuning methods, whose effect lies primarily in feature adaptation. Making this hyperparameter trainable gives rise to a novel and highly parameter-efficient finetuning method. Empirically, CT improves both generalization and robustness. For example, it boosts downstream accuracy of ResNet-50/152 by 8.59%/8.34% over linear probing and 4.64%/1.70% over LoRA across 12 datasets, and improves robust accuracy on the $\ell_\infty$ benchmark from RobustBench by 1032.64%/1494.46%. Our code is available at https://github.com/Leon-Leyang/curvature-tuning.

## 1 Introduction

The scaling of model and data sizes has given rise to foundation models, such as Llama3 [1] for natural language processing (NLP), DINOv2 [2] for computer vision (CV), CLIP [3] and SigLIP [4] for multimodal tasks, and OpenVLA [5] for embodied agents. These models have shown remarkable capabilities, accelerating a paradigm shift in artificial intelligence: transitioning from training task-specific models from scratch to leveraging models pretrained on large datasets and finetuning them for downstream applications.

Full finetuning, the process of steering[1] a pretrained model by adapting all its parameters to downstream datasets, was once the primary approach for transferring knowledge. While it effectively enhances generalization [6] and robustness [7], it is computationally expensive at large model scales. To mitigate this, parameter-efficient finetuning (PEFT) methods such as Serial Adapter [8] and LoRA [9] have been introduced, which finetune only a small subset of parameters. However, these approaches usually lack interpretability and principled design. For instance, they treat the model as a black box, making it unclear how the model is steered for downstream tasks. Consequently, they usually rely on heuristic choices—such as LoRA's rank, placement, and initialization—with minimal

---

[*]Equal contribution.

[†]Work done at Brown University.

[1]We use *steering* as a general term for tuning a model, including training- and non-training-based methods. We use *finetuning* to refer specifically to steering methods that adapt the model's parameters via training.

39th Conference on Neural Information Processing Systems (NeurIPS 2025).

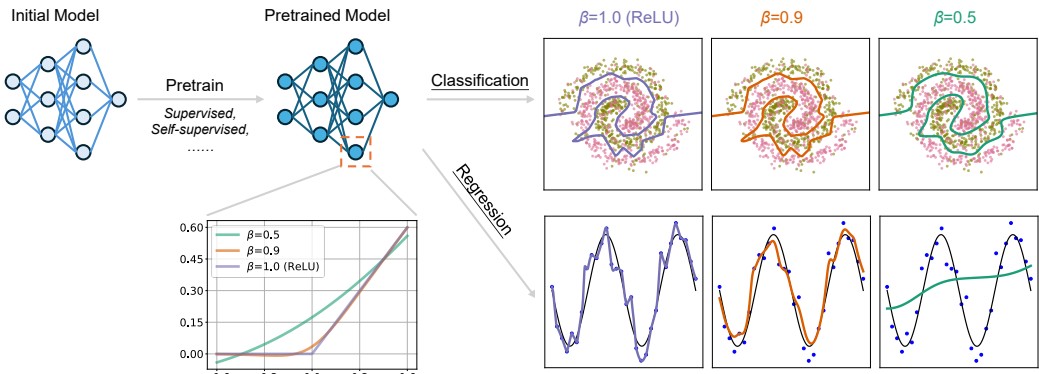

Figure 1: Illustration of Curvature Tuning (CT) on classification (top) and regression (bottom) tasks. The pretrained model for classification is a 3-layer MLP with hidden width 20 trained for 2000 steps; for regression, it is a 9-layer MLP with hidden width 64 trained for 20000 steps. **CT steers a pretrained model by replacing ReLUs with a $\beta$-parameterized activation function and tuning $\beta$ from 1 to 0, effectively modulating the model's decision boundary curvature.**

theoretical guidance. This leads to a natural question: *how can we construct principled steering solutions addressing both efficiency and interpretability?*

In this work, we answer the question by introducing a novel model steering perspective. We observe that despite differences in specific forms, existing finetuning methods all share a focus on adapting model weights—either by introducing new ones or updating existing ones. However, one critical component of the model has been largely overlooked: the activation functions (e.g., ReLU), which are responsible for the model's nonlinearity and, ultimately, its expressivity [10, 11].

Grounded in the spline interpretation of deep networks [12, 13], we propose **Curvature Tuning (CT)**, a steering method (denoted as S-CT) that provably modulates a model's decision boundary curvature by injecting a single hyperparameter $\beta$ into the activation function, as shown in Fig. 1. Additionally, allowing $\beta$ to be trained leads to a novel finetuning method (denoted as T-CT). We highlight four key advantages of CT below:

**CT is more interpretable and principled.** We show that CT provably modulates the curvature of the model's decision boundary with as few as only one hyperparameter.

**CT complements current finetuning methods.** More essentially, while current finetuning methods adapt features, CT projects the model to a space of smooth functions.

**CT is highly parameter-efficient.** As a steering method, S-CT introduces only one (hyper)parameter per network. As a finetuning method, T-CT still uses significantly fewer parameters than LoRA with rank one, requiring only 0.58% to 59.09% of the parameters used by LoRA in our experiments.

**CT improves both generalization and robustness.** For example, T-CT boosts transfer accuracy of ResNet-50/152 by 8.59%/8.34% over linear probing and 4.64%/1.70% over LoRA with rank one across 12 downstream datasets. S-CT improves robust accuracy of ResNet-50/152 by 1032.64%/1494.46% on the $\ell_\infty$ benchmark from RobustBench.

In summary, our key contributions are both theoretical and empirical. **Theoretically**, we propose CT and show that it provably modulates the model's decision boundary curvature, by projecting the model onto a space of smooth functions. **Empirically**, we introduce S-CT as a steering method and T-CT as a finetuning method, demonstrating improved generalization across six models and 12 downstream datasets, as well as robustness gains on the RobustBench benchmark [14].

The remainder of this paper is organized as follows: Section 2 reviews current finetuning techniques and introduces relevant spline concepts, the foundation for our method. Section 3 details our proposed method and its theoretical guarantees. Section 4 presents experimental results, and Section 5 summarizes our findings and potential future directions.

## 2 Background

This section first reviews current finetuning techniques and their limitations (Section 2.1), then introduces spline theory and its connections to deep networks as the foundation for CT (Section 2.2).

### 2.1 Current finetuning techniques and limitations

Finetuning refers to steering a pretrained model to improve its performance on downstream tasks through training. Initially, the common practice was to continue training all model parameters—a process known as *full finetuning*. Notable examples include GPT [6] and DINO [15]. However, as model sizes have grown, full finetuning has become increasingly costly and often impractical, particularly when downstream datasets are small. Given these challenges, *parameter-efficient finetuning (PEFT)* methods were developed to mitigate the cost while maintaining effectiveness.

We follow the categorization of Han et al. [16], which groups PEFT methods into four main categories. **Additive PEFT** introduces additional trainable parameters to the pretrained model, training only these new parameters during finetuning. Examples include Serial Adapter [8], Prefix-tuning [17], (IA)$^3$ [18], and RoAd [19]. **Selective PEFT** identifies a subset of existing parameters for finetuning, with examples such as U-Diff pruning and S-Diff pruning [20]. **Reparameterized PEFT** decomposes pretrained weights into low-rank matrices, finetuning only the low-rank components, which are converted back during inference; examples include LoRA [9] and DyLoRA [21]. **Hybrid PEFT** combines multiple PEFT strategies, such as UniPELT [22] and S4 [23].

While PEFT methods differ in the parameters they update, they all adapt model weights and operate on learned features—an approach that often relies on heuristic tuning. For example, LoRA requires decisions about adapter placement [24], rank [21, 25], scaling [26], and initialization [27]. In contrast, as described in Section 3, CT introduces only a single hyperparameter into the activation functions that provably modulates the decision boundary curvature, offering a more interpretable alternative that instead operates on the model's underlying function space without changing model weights.

The problem of learning low-curvature activation functions has been explored in the context of pretraining, where Srinivas et al. [28] exploit a parametric version of Softplus. Our paper differs in two substantial aspects. First, we introduce a more general parametric activation function (CTU) encompassing a wider family of non-linearities, which provides direct applicability to current network architectures and particularly transformers. Second, we are the first to propose modulating activation functions as a novel PEFT direction, allowing to control downstream generalization and robustness without changing the pre-trained model's learned features.

### 2.2 The spline formulation of deep networks

In this subsection, we review relevant concepts in splines, which provide a mathematical framework for understanding the relationship between piecewise-affine functions and deep networks (DN).

A *spline function* is a continuous function $s : \mathbb{R}^D \to \mathbb{R}$ defined piecewise by polynomials. An *affine spline function* is a special case where each piece is defined by an affine mapping. Such a function can be parameterized by three components: a matrix $\mathbf{A} \in \mathbb{R}^{R \times D}$ representing the slopes of the affine mappings, a vector $\mathbf{b} \in \mathbb{R}^R$ representing the offsets, and a partition $\Omega \triangleq \{\omega_1, \ldots, \omega_R\}$ of the input space $\mathbb{R}^D$ into $R$ regions. For an input $\mathbf{x} \in \mathbb{R}^D$, the affine spline function is defined as:

$$s[\mathbf{A}, \mathbf{b}, \Omega](\mathbf{x}) = \sum_{r=1}^{R} \left( \langle \mathbf{A}_{r,\cdot}, \mathbf{x} \rangle + \mathbf{b}_r \right) \mathbf{1}_{\{\mathbf{x} \in \omega_r\}}, \tag{1}$$

where $\mathbf{1}_{\{\mathbf{x} \in \omega_r\}}$ is an indicator function that equals 1 if $\mathbf{x}$ belongs to region $\omega_r$ and 0 otherwise.

A *max-affine spline function* is a special case of an affine spline function that does not need explicit knowledge of $\Omega$. Instead, its output is computed as the maximum value over the affine mappings:

$$s[\mathbf{A}, \mathbf{b}](\mathbf{x}) = \max_{r=1...R} \left( \langle \mathbf{A}_{r,\cdot}, \mathbf{x} \rangle + \mathbf{b}_r \right). \tag{2}$$

The key result underpinning our study is that many DN layers—such as fully connected and convolutional layers, and convex piecewise-affine activations (e.g., ReLU, max pooling, and maxout)—can

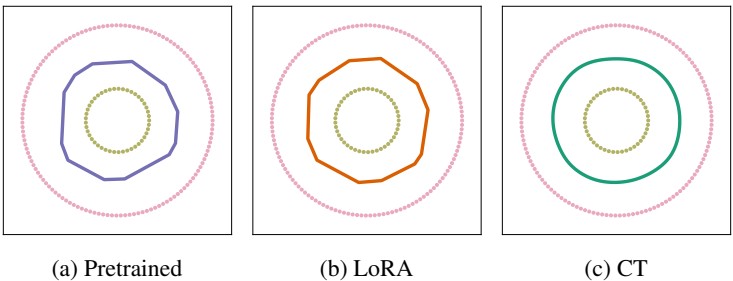

|      (a) Pretrained      |      (b) LoRA      |      (c) CT      |

Figure 2: Toy example illustrating how modulating model's activation functions steers decision boundaries curvature. The model is a 2-layer MLP with hidden width 7; (a) baseline pretrained for 4000 steps, then finetuned for another 4000 steps using (b) LoRA ($r = 1$, $\alpha = 1$) and (c) *Trainable CT*. **CT achieves near-optimal approximation by smoothing the decision boundary of the pretrained model, whereas LoRA only operates on the model parameters, without changing the model's underlying geometry.**

be exactly represented as max-affine spline operators[2], and that DNs composed of such layers can be represented as affine spline operators [13] (further details in Appendix A).

Now that we have reviewed existing finetuning methods and their limitations, and introduced the necessary spline-based foundations, we proceed to present our proposed method in Section 3.

## 3  Curvature Tuning (CT): a provable method for model steering

In this section, we introduce our proposed method, Curvature Tuning (CT). We begin by motivating the benefits of modulating decision boundaries as a model steering technique. Then, we dive into CT's construction in Section 3.1 with implementation details in Section 3.2. Additional theoretical intuition is provided in Section 3.3. Extensive experimental validation is presented in Section 4.

**Motivating example.**    Consider a toy binary classification problem in $\mathbb{R}^2$, whereby the optimal decision boundary separating two classes is given by the unit circle $S^1 = \{\mathbf{x} \in \mathbb{R}^2 : \|\mathbf{x}\|_2 = 1\}$, parameterized by the curve $\gamma : t \mapsto (\cos 2\pi t, \sin 2\pi t)$, for $t \in [0, 1]$. Let $\sigma(z) = \frac{\exp(z)}{1+\exp(z)}$ be the sigmoid. By definition of decision boundary, an optimal network $f : \mathbb{R}^2 \to \mathbb{R}$ should predict both classes with equal probability $\sigma(f(\gamma(t))) = 0.5, \forall t \in [0, 1] \iff f(\gamma(t)) = 0, \forall t$. Focusing on the decision boundary, the approximation error $e$ is given by the line integral $e = \int_\gamma |f(\mathbf{x})|d\mathbf{x} = \int_0^1 |f(\gamma(t))| \|\gamma'(t)\| dt$. For a ReLU network (Eq. (1)), computing the error over the regions $\Omega_\gamma = \Omega \cap S^1$ yields $e = 2\pi \sum_{k=0}^{r'} \left( (-1)^{z_k(t_k)} [g_{r_k}(t)]_{t_k}^{s_k} + (-1)^{z_k(t_{k+1})} [g_{r_k}(t)]_{s_k}^{t_{k+1}} \right)$ for $r' = |\Omega_\gamma|$, where $t_k$ are the spline breakpoints pulled-back from $\mathbb{R}^2$ to $[0, 1]$, $r_k$ denotes which spline region the $k$-th segment $[t_k, t_{k+1}]$ falls into, $z_k(t) := \mathbf{1}_{\{\mathbf{A}_{r_k,\cdot}\gamma(t)+\mathbf{b}_{r_k}<0\}}$, $s_k \in [t_k, t_{k+1}]$ is the value flip point of $z_k(t)$ and $g_{r_k}(t) = \mathbf{A}_{r_k,1}\frac{\sin 2\pi t}{2\pi} - \mathbf{A}_{r_k,2}\frac{\cos 2\pi t}{2\pi} + \mathbf{b}_{r_k}t$ (full derivations in Appendix C.2). Assuming the network considered attains optimal approximation error $e > 0$, then it is clear that $t_{k+1} \to t_k \implies e \to 0$. The converse implication holds when the task is changed from classification to regressing the unit circle. In this case, reducing the approximation error by adapting the model's weights (either through PEFT or training a larger model from scratch) will still result in an affine spline operator, for which $e > 0$ and further reducing $e$ to 0 would require infinitely many neurons. This paper explores an orthogonal approach: by modulating the model's activation functions, one can efficiently control its curvature and, in turn, that of its decision boundaries, thereby steering them toward optimality—without modifying the model's weights. Fig. 2 illustrates this phenomenon: methods such as LoRA implicitly tune the spline slopes and breakpoints, whereas modulating model nonlinearities changes the model's underlying geometry.

---

[2]Here, *operator* simply refers to a mapping between vector spaces (i.e., a function $f : \mathbb{R}^D \to \mathbb{R}^K$) obtained by concatenating $K$ functions like $s$.

### 3.1 The $\beta$-Vector-Quantization (VQ) inference framework

This section builds upon the max-affine spline formulation from Eq. (2) to construct a model steering method operating on the model's activation functions. By inspecting Eq. (2), we observe that the mapping remains affine within each (implicitly defined) region where the pointwise maximum does not change. Specifically, for any input $\mathbf{x}$ where $\arg\max_{r=1\ldots R}\left(\langle\mathbf{A}_{r,\cdot},\mathbf{x}\rangle+\mathbf{b}_r\right)$ remains constant, all such inputs belong to the same region, as they share the same affine mapping. The nonlinearity of the function arises when transitioning between these regions.

**Smoothing the nonlinearity by smoothing the spline region assignment process.** Instead of going from one affine mapping to another in an abrupt fashion (whenever crossing that hyperplane), one may consider a smoother transition. There are two common practices to achieve that goal.

We know that each unit of a layer is a max-affine spline. The inference process of each unit can thus be decomposed into two steps:
**1. VQ Inference Step (region selection)**: Determine the affine transformation that maximizes the output, which can be viewed as a VQ process. The decision is encoded in a one-hot selection variable $\mathbf{t} \in \mathbb{R}^R$, where $R$ is the number of input region partitions of the max-affine spline function, and the $r^*$-th entry is set to 1, where:

$$r^* = \arg\max_{r\in\{1,\ldots,R\}}\left(\langle\mathbf{A}_{r,\cdot},\mathbf{x}\rangle+\mathbf{b}_r\right). \tag{3}$$

**2. Computation Step (affine transformation)**: Compute the output of the neuron based on the selection variable $\mathbf{t}$:

$$f(\mathbf{x}) = \sum_{r=1}^{R}\mathbf{t}_r \cdot \left(\langle\mathbf{A}_{r,\cdot},\mathbf{x}\rangle+\mathbf{b}_r\right). \tag{4}$$

As discussed, the affine transformation is selected in a *hard* manner, where only the transformation that maximizes the output is chosen. Alternatively, a *soft* selection can be employed, in which the selection variable $\mathbf{t}$ is no longer a one-hot vector but is inferred probabilistically. To formalize this, we follow the probabilistic formulation from [29] and introduce the following regularized region selection problem, where the new selection variable $\mathbf{t}^\beta$ is computed as below:

$$\mathbf{t}^\beta = \arg\max_{\mathbf{t}\in\Delta_R}\left[\beta\sum_{r=1}^{R}\mathbf{t}_r \cdot (\langle\mathbf{A}_{r,\cdot},\mathbf{x}\rangle+\mathbf{b}_r) + (1-\beta)H(\mathbf{t})\right], \tag{5}$$

where $H(\mathbf{t})$ denotes the Shannon entropy of the selection variable, and $\Delta_R$ is the probability simplex in $\mathbb{R}^R$. The optimal solution $\mathbf{t}^\beta$ has the closed-form:

$$\mathbf{t}_r^\beta = \frac{\exp\left(\frac{\beta(\langle\mathbf{A}_{r,\cdot},\mathbf{x}\rangle+\mathbf{b}_r)}{1-\beta}\right)}{\sum_{i=1}^{R}\exp\left(\frac{\beta(\langle\mathbf{A}_{i,\cdot},\mathbf{x}\rangle+\mathbf{b}_i)}{1-\beta}\right)} \quad \text{for } r = 1,\ldots,R. \tag{6}$$

Using the computation step in Eq. (4) and a ReLU activation function, switching from $\beta = 1$ to $\beta = 0.5$ is provably equivalent to replacing ReLU with the Sigmoid Linear Unit (SiLU). In the limit as $\beta \to 0$, the activation function becomes linear—thus making the entire input-output mapping of the network linear as well. [29]

**Smoothing the nonlinearity by smoothing the max.** As previously mentioned, there is an alternative way to smooth the max-affine spline mapping from Eq. (2). Instead of relying on a soft region assignment, we can instead directly smooth the maximum function. It is already well known that smoothing the maximum operator leads to the log-sum-exp operator (i.e. Softplus). Hence, the mapping from Eq. (2) now becomes

$$(1-\beta)\ln\left[\sum_{r=1}^{R}\exp\left(\frac{\langle\mathbf{A}_{r,\cdot},\mathbf{x}\rangle+\mathbf{b}_r}{1-\beta}\right)\right], \tag{7}$$

where we parameterized the mapping so that its behavior is akin to Eq. (5), a value of $\beta \to 1$ recovers the original affine spline activation, e.g., ReLU.

The crucial observation we make is that both parameterizations tend to shift the mean of the output of the unit either by a negative factor (for Eq. (5)) or a positive factor (for Eq. (7)). This means

that in very deep models, varying $\beta$ with either parameterization produces a shift in the decision boundary or regression that cannot be recovered unless the parameters are trained once again, which we are trying to avoid. As a result, as detailed in Section 3.2, our implementation combines the two parameterizations in a weighted manner to mitigate this bias, as Fig. S1 illustrates.

## 3.2 Implementation of CT

We begin by presenting the core activation that gives CT its expressive power—referred to as **CT Unit (CTU)**. The activation is obtained by combining the two parameterizations discussed in Section 3.1, in order to mitigate the mean shift introduced by each parameterization individually:

$$\varphi_{\beta,c}(\mathbf{x}) = c \cdot \sigma \left( \frac{\beta \mathbf{x}}{1-\beta} \right) \cdot \mathbf{x} + (1-c) \cdot \ln \left[ 1 + \exp \left( \frac{\mathbf{x}}{1-\beta} \right) \right] \cdot (1-\beta), \qquad (8)$$

where $\beta \in [0,1]^3$ modulates the curvature, $c \in [0,1]$ is the mixing coefficient, and $\sigma(\cdot)$ denotes the sigmoid function. This is essentially a convex combination of reparameterized SiLU and Softplus:

$$\text{SiLU}(\mathbf{x}) = \sigma(\eta \mathbf{x}) \cdot \mathbf{x}, \ \eta = \frac{\beta}{1-\beta}; \quad \text{Softplus}(\mathbf{x}) = \frac{1}{\gamma} \cdot \ln \left[ 1 + \exp \left( \gamma \mathbf{x} \right) \right], \ \gamma = \frac{1}{1-\beta}. \quad (9)$$

It is worth noting that CTU naturally encompasses a broad family of functions. Particularly, it recovers SiLU ($c = 1$), Softplus ($c = 0$), as well as closely approximates GELU ($c = 1$ and $\beta = 0.64$).

**Steering vs Trainable CT**. We provide two implementations of CT differing in how CTU is applied. The first, denoted *Steering CT* (S-CT), replaces all ReLUs in the network with CTUs using a fixed $c = 0.5$ and a shared $\beta \in [0,1]$. This version is highly parameter-efficient—introducing only a single hyperparameter—and does not require backpropagation, making it suitable as a steering method.

The second, referred to as *Trainable CT* (T-CT), also replaces all ReLUs with CTUs but assigns each output neuron its own trainable pair $(\beta, c)$, optimized via backpropagation. This version serves as a finetuning method: while it introduces additional parameters, the increase is modest compared to methods like LoRA and it yields competitive performance, as shown in Section 4.2. Code for both implementations is provided in Appendix D.

## 3.3 Curvature Tuning operates as a projection

This section provides a characterization of CT, by casting it as a projection of a ReLU network to a space of smooth functions. All proofs are deferred to Appendix C.1.

**Theorem 3.1** (Informal). *For a ReLU network $f : \mathbb{R}^D \to \mathbb{R}$ with parameter $\mathbf{W}$ (collecting all weights and biases), for fixed $c \in [0,1]$ and $\beta \in [0,1)$, replacing every instance of ReLU with a CTU (Eq. (8) with hyperparameters $\beta, c$ is equivalent to projecting $f$ to a smooth function $f_{\beta,c}$ with bounded gradients and curvature, while keeping $\mathbf{W}$ fixed. Importantly, for $0 < \beta < 1$, $f_{\beta,c}$ enjoys higher local expressivity than $f$ for the same parameter $\mathbf{W}$, due to non-vanishing local curvature.*

To conclude, we observe how varying $\beta$ modulates the curvature of the whole model function $f$ and, in turn, of the model's decision boundaries. We begin by noting that for a deep network $f : \mathbb{R}^D \to \mathbb{R}^K$, the decision boundary between any class $i$ and $j$ is given by $\{\mathbf{x} \in \mathbb{R}^D : g(\mathbf{x}) := f_i(\mathbf{x}) - f_j(\mathbf{x}) = 0\}$, for any $i, j = 1, \ldots, K$ with $i \neq j$. Particularly, $g$ is itself a deep network, sharing the same parameters as $f$ up until the penultimate layer, after which the parameters are the vector $W_i^L - W_j^L$ and the bias $\mathbf{b}_i^L - \mathbf{b}_j^L$. Importantly, *when varying $\beta$ while keeping all model parameters fixed*, the Jacobian $\nabla_{\mathbf{x}} g(\mathbf{x})$ and the Hessian $\nabla_{\mathbf{x}}^2 g(\mathbf{x})$ are respectively given by the gradients and Hessian of $\mathbf{z}^{L-1}(\mathbf{x})$ – corresponding to the post-activation output of the $L-1$-th layer—weighted by $W_i^L - W_j^L$. Hence, modulating the nonlinearity of activation functions via $\beta$ directly controls the curvature of both model function and its decision boundaries.[4]

Particularly, for $c = 1$ (Eq. (8)), as $\beta \to 0$, the activation becomes linear. Since modern DNs (e.g. MLP, CNN, RNN) are composed of activation functions interleaved with affine layers, it follows

---

[3]In practice, for numerical stability we use $\eta = \frac{\beta}{1-\beta+\varepsilon}$ and $\gamma = \frac{1}{1-\beta+\varepsilon}$, where $\varepsilon = 10^{-6}$ allows the method to remain well-defined at $\beta = 1$.

[4]In the following, unless specified, we will thus refer interchangeably to the curvature of a DN mapping and that of its decision boundaries whenever modulating nonlinearities via CT.

directly that the entire input-output mapping becomes affine when $\beta \to 0$. In this setting, the curvature of the mapping—defined as the norm of its Hessian—becomes zero. As a result, transitioning from the original DN mapping ($\beta = 1$) to the linear setting effectively modulates the network decision boundary curvature, reducing it continuously to zero in the limit. For $c < 1$, as $\beta \to 0$, the model retains non-vanishing local curvature, while the mapping becomes smooth.

## 4 Enhancing model generalization and robustness with CT

In this section, we empirically validate the effectiveness of CT across multiple settings. We first demonstrate that S-CT improves generalization (Section 4.1), while T-CT achieves improvement comparable to LoRA with substantially fewer parameters (Section 4.2). Then we show that CT can improve the robustness of models through its implicit bias (Section 4.3). Finally, we demonstrate CT's effectiveness on transformers despite only partial guarantees (Section 4.4). GPU and seed details are provided in Appendix B.

### 4.1 Improving generalization on downstream datasets with S-CT

In this subsection, we evaluate the effectiveness of S-CT in improving model generalization on a variety of downstream datasets. Specifically, we transfer ImageNet-pretrained ResNet-18, ResNet-50, ResNet-152 and VGG-11 models to 12 downstream tasks, including Arabic Characters [30], Arabic Digits [31], Beans [32], CUB-200-2011 [33], DTD [34], FashionMNIST [35], FGVC-Aircraft [36], Flowers102 [37], Food101 [38], and three subsets from MedMNIST—PathMNIST, OCTMNIST, and DermaMNIST [39]. Each dataset is split into training/validation/test sets (details in Appendix B.1).

To apply S-CT, we replace all ReLUs in the backbone with CTUs, freeze the model, and train a linear classifier on the penultimate layer. The optimal $\beta$ is selected via grid search over $\beta \in [0.7, 1]$ (step size 0.01) using validation accuracy, and the corresponding test accuracy is reported. For the baseline, we train a linear classifier on the frozen original model and report the test accuracy of the checkpoint that performs best on the validation set. All linear classifiers use the same training setup detailed in Appendix B.1.

Table 1: Mean accuracy (%) over three runs of ImageNet-pretrained ResNet-18/50 when transferred to 12 downstream datasets. The second row under each method indicates the number of trainable parameters (excluding the linear classifier). **S-CT outperforms linear probing on the frozen backbone, and T-CT surpasses LoRA (rank 1).** Full results (± std) in Table S1.

| Dataset | ResNet-18 | | | | ResNet-50 | | | |
|---|---|---|---|---|---|---|---|---|
| | Frozen (0) | S-CT (1) | LoRA (35923) | T-CT (3968) | Frozen (0) | S-CT (1) | LoRA (79443) | T-CT (45440) |
| Arabic Characters | 81.91 | 87.65 | 93.37 | **93.76** | 80.65 | 83.66 | 94.21 | **95.67** |
| Arabic Digits | 97.93 | 98.77 | **99.08** | 99.03 | 98.33 | 98.37 | 99.08 | **99.16** |
| Beans | 87.76 | 90.36 | 93.23 | **94.01** | 89.58 | 91.93 | 94.79 | **95.57** |
| CUB-200 | 62.84 | 63.18 | 54.83 | **64.30** | 65.23 | 64.62 | 66.17 | **71.03** |
| DTD | 62.80 | 62.66 | 54.36 | **63.62** | **67.34** | 66.91 | 64.70 | 65.07 |
| FashionMNIST | 88.63 | 88.70 | **91.65** | 91.07 | 90.05 | 90.34 | 92.19 | **92.78** |
| FGVC-Aircraft | 36.80 | 38.68 | 29.19 | **46.44** | 38.03 | 41.16 | 41.99 | **55.70** |
| Flowers102 | 80.86 | 81.97 | 67.53 | **86.55** | 84.00 | 83.84 | 82.58 | **87.62** |
| Food101 | 61.41 | 62.27 | 64.40 | **66.04** | 68.06 | 68.02 | 71.42 | **73.60** |
| DermaMNIST | 74.83 | 75.05 | 74.21 | **77.66** | 75.94 | 75.89 | 75.73 | **78.02** |
| OCTMNIST | 65.03 | 67.27 | **74.27** | 69.53 | 67.53 | 68.00 | **75.90** | 74.13 |
| PathMNIST | 86.77 | 87.51 | **87.62** | 87.17 | 90.08 | **90.26** | 85.43 | 87.33 |
| *Average* | 73.96 | 75.34 | 73.64 | **78.26** | 76.24 | 76.92 | 78.68 | **81.31** |

Tables 1 and S1 show mean accuracy over three runs for linear probing with and without S-CT on 12 downstream datasets using ResNet-18/50/152 and VGG-11 backbones. S-CT consistently improves generalization, with average relative gains of 1.97%, 1.16%, 0.02%, and 0.71% respectively. We also report the average optimal $\beta$ across datasets: 0.84 for ResNet-18, 0.94 for ResNet-50, 0.96 for ResNet-152 and 0.90 for VGG-11 (full results in Table S2). These values are consistently close to 1,

suggesting the search range can be narrowed for efficiency. Example accuracy curves in Fig. S2 show that accuracy varies smoothly with $\beta$ and typically peaks in the middle of the search range.

Additionally, we conduct ablation experiments on the choice of $c = 0.5$ for S-CT under the same settings. As shown in Table S3, setting $c = 0.5$ yields better performance than $c = 1$ (SiLU) or $c = 0$ (Softplus) by 1.99% and 0.56% respectively on ResNet-18, and by 0.94% and 0.57% on ResNet-50, while performing slightly worse by 0.05% and 0.73% on ResNet-152.

## 4.2 T-CT is comparable to LoRA with fewer parameters

In this subsection, we show that T-CT further improves generalization, achieving performance comparable to LoRA with fewer parameters. We conduct experiments using the same setup as in Section 4.1, evaluating both T-CT and LoRA. For both methods, we replace the original linear classifier in each model with an appropriate one, and apply the respective method to the pretrained backbone. In T-CT, we initialize all $\beta$ parameters to 0.8 and all $c$ parameters to 0.5. For LoRA, we apply it to all convolutional and linear layers in the backbone (implementation details in Appendix E). And we set the rank $r = 1$ and scale $\alpha = 1$, so that it has a comparable number of parameters to T-CT. Refer to Appendix B.2 for training configurations applied to the two methods. Here we report the test accuracy of the best checkpoint on the validation set.

The results, summarized in Tables 1 and S1, show that T-CT achieves the best performance across all methods, with average relative improvements on ResNet-18/50/152 and VGG-11 of 6.75%, 8.59%, 8.34% and 5.53% over the baseline; 4.62%, 7.13%, 8.35% and 4.73% over S-CT; and 10.20%, 4.64%, 1.70% and 4.05% over LoRA. Importantly, T-CT achieves better performance than LoRA with fewer parameters. As reported in Tables 1 and S1, the number of trainable parameters (excluding the classifier) used by T-CT amounts to only 11.05%, 57.20%, 59.09% and 36.39% of that used by LoRA on the four models respectively—even with LoRA operating at its lowest-rank setting ($r = 1$). This highlights the parameter efficiency of our approach. We further compare T-CT and LoRA with varying ranks ($r \in \{1, 2, 4\}$) and scaling factors ($\alpha \in \{r, 2r, 4r\}$) in Table S4. For each dataset and LoRA rank, we report the best test accuracy achieved among the candidate scaling factors. As shown, T-CT can still outperform LoRA under this more challenging setting, achieving relative improvements of 9.65%, 6.56%, and 3.34% over the three ranks on ResNet-18; 4.18%, 2.27%, and 0.19% on ResNet-50; and 1.22%, –0.91%, and –1.84% on ResNet-152, further demonstrating the effectiveness and parameter efficiency of T-CT.

To better understand how T-CT behaves during training, we analyze the distributions of learned $\beta$ and $c$ values (full statistics provided in Tables S5 and S6). We observe a high degree of within-model variation, with standard deviations ranging from 0.31 to 0.38, while the means remain remarkably stable across architectures: 0.69 to 0.79 for $\beta$ and 0.57 to 0.61 for $c$. These mean values are close to those used in S-CT, though the learned $\beta$ values tend to be smaller than the optimal shared $\beta$ found in S-CT (0.84 to 0.96), while the learned $c$ values are larger than the fixed $c = 0.5$. We further visualize the distributions in Figs. S3 and S4. In most datasets, as shown in Fig. S3 (OCTMNIST), both $\beta$ and $c$ exhibit a sharp U-shaped distribution—concentrating near 0 and 1 with a flat middle. This suggests that T-CT leverages its parameter flexibility to assign values at the extremes, producing an effective average close to the manually chosen settings in S-CT, rather than concentrating around the mean values themselves.[5] In a few datasets, we observe deviations from this trend, as exemplified in Fig. S4 (DTD). Nonetheless, a consistent pattern is that for any given dataset, the distributions remain visually similar across models.

## 4.3 Improving model robustness through CT's implicit bias

In this subsection, we demonstrate that CT exhibits an implicit bias toward enhancing model robustness, using benchmarks from RobustBench [14]. We evaluate the robustness of ResNet-18/50/152 on CIFAR-10/100 and ImageNet using the official $\ell_2$ and $\ell_\infty$ adversarial benchmarks from RobustBench [14] and the common corruption benchmark [40]. See Appendix B.3 for more details.

More specifically, we first show that S-CT can improve robustness without any robustness-oriented objective, by applying it to pretrained models, performing a grid search over $\beta \in [0.7, 1]$ with a step

---

[5]This behavior may in part be influenced by the sigmoid-based parameterization used in our implementation of T-CT to constrain $\beta$ and $c$ during training.

size of 0.01, and reporting the best robust accuracy achieved. We then show that T-CT, when used for finetuning (from ImageNet to CIFAR-10/100 following the same training configurations as in Sections 4.1 and 4.2), also enhances robustness despite the absence of a robustness objective—the finetuned models achieve stronger robustness than those trained with linear probing or LoRA ($r = 1$).

Table 2: Mean robust accuracy (%) over three runs of ImageNet-pretrained ResNet-18/50/152 under $\ell_2/\ell_\infty$ attacks and corruptions on CIFAR-10/100 and ImageNet. **S-CT yields substantial improvements under $\ell_\infty$ attacks, with the selected $\beta$ values close to 1.** Full results (± std) in Table S7.

| Model | Dataset | $\ell_2$ | | | $\ell_\infty$ | | | Corruption | | |
| | | Frozen | S-CT | $\beta$ | Frozen | S-CT | $\beta$ | Frozen | S-CT | $\beta$ |
|---|---|---|---|---|---|---|---|---|---|---|
| ResNet18 | CIFAR10 | 53.67 | 53.67 | 1.00 | 11.17 | **14.93** | 0.90 | 77.73 | 77.73 | 1.00 |
| | CIFAR100 | 24.30 | **25.50** | 0.92 | 4.47 | **6.90** | 0.92 | 51.81 | **51.95** | 0.94 |
| | ImageNet | 23.37 | 23.37 | 1.00 | 0.00 | **7.00** | 0.89 | 33.11 | **33.32** | 0.92 |
| | *Average* | 33.78 | **34.18** | 0.97 | 5.21 | **9.61** | 0.90 | 54.22 | **54.33** | 0.95 |
| ResNet50 | CIFAR10 | 55.10 | **56.53** | 0.97 | 10.10 | **12.08** | 0.90 | 77.26 | 77.26 | 1.00 |
| | CIFAR100 | 23.83 | **25.80** | 0.96 | 4.43 | **7.90** | 0.93 | 53.91 | **53.93** | 0.98 |
| | ImageNet | 31.90 | 31.90 | 1.00 | 0.30 | **9.30** | 0.93 | 39.64 | 39.64 | 1.00 |
| | *Average* | 36.94 | **38.08** | 0.98 | 4.94 | **9.76** | 0.94 | 56.94 | 56.94 | 0.99 |
| ResNet152 | CIFAR10 | 56.27 | 56.27 | 1.00 | 11.47 | **15.00** | 0.99 | 78.82 | **78.83** | 0.99 |
| | CIFAR100 | 27.90 | **28.23** | 0.98 | 5.40 | **7.70** | 0.99 | 56.12 | 56.12 | 1.00 |
| | ImageNet | 42.50 | 42.50 | 1.00 | 0.30 | **13.53** | 0.97 | 45.47 | 45.47 | 0.99 |
| | *Average* | 42.22 | **42.33** | 0.99 | 5.72 | **12.08** | 0.98 | 60.14 | 60.14 | 0.99 |

Table 3: Mean robust accuracy (%) over three runs of ImageNet-pretrained ResNet-18/50/152 transferred to CIFAR-10/100 under $\ell_2$, $\ell_\infty$ attacks, and corruptions. **T-CT improves $\ell_\infty$ robustness significantly compared to linear probing and LoRA.** Full results (± std) in Table S8.

| Model | Dataset | $\ell_2$ | | | $\ell_\infty$ | | | Corruption | | |
| | | Frozen | LoRA | T-CT | Frozen | LoRA | T-CT | Frozen | LoRA | T-CT |
|---|---|---|---|---|---|---|---|---|---|---|
| ResNet18 | CIFAR10 | 8.47 | 5.93 | **8.93** | 0.30 | 0.70 | **1.57** | **21.34** | 13.59 | 16.83 |
| | CIFAR100 | **1.57** | 0.77 | 1.10 | 0.03 | 0.07 | **0.17** | **5.10** | 2.96 | 4.62 |
| | *Average* | **5.02** | 3.35 | 5.01 | 0.16 | 0.38 | **0.87** | **13.22** | 8.28 | 10.72 |
| ResNet50 | CIFAR10 | 6.23 | 4.57 | **6.83** | 0.20 | 0.33 | **2.43** | **16.23** | 11.69 | 12.68 |
| | CIFAR100 | **0.70** | 0.37 | 0.47 | 0.00 | 0.03 | **0.07** | **3.47** | 2.04 | 1.61 |
| | *Average* | 3.47 | 2.47 | **3.65** | 0.10 | 0.18 | **1.25** | **9.85** | 6.86 | 7.14 |
| ResNet152 | CIFAR10 | **8.03** | 4.63 | 8.00 | 0.43 | 0.20 | **5.10** | 13.82 | 11.33 | 9.83 |
| | CIFAR100 | **0.90** | 0.47 | 0.50 | **0.17** | 0.00 | 0.00 | 2.07 | **2.13** | 1.72 |
| | *Average* | **4.46** | 2.55 | 4.25 | 0.30 | 0.10 | **2.55** | **7.94** | 6.73 | 5.78 |

For S-CT, as summarized in Table 2, it is particularly effective against $\ell_\infty$ attacks, achieving substantial relative improvements of 44.01%, 1032.64%, and 1494.46% for ResNet-18, ResNet-50, and ResNet-152.[6] Improvements under $\ell_2$ attacks and common corruptions are comparatively moderate. The corresponding optimal $\beta$ values, also shown in Table 2, are consistently close to 1, suggesting that modest curvature modulation suffices to improve robustness—further highlighting the practical efficiency of S-CT. For T-CT, as shown in Table 3, it likewise enhances $\ell_\infty$ robustness significantly, achieving average relative improvements over linear probing of 445.00%, 1115.00%, and 493.02%, and over LoRA of 133.57%, 384.85%, and 2450.00%. These results collectively demonstrate CT's implicit bias toward robustness enhancement, further validating its effectiveness beyond standard generalization improvements.

---

[6]We exclude from relative-improvement computation any cases where the baseline robust accuracy is 0.

## 4.4 CT shows promise on transformers

In this subsection, we investigate the effectiveness of T-CT in improving the generalization of transformer architectures. Unlike ResNets, transformers include attention layers that fall outside the max-affine spline framework and typically employ non-ReLU activation functions (e.g., GELU), which weakens our theoretical guarantees.

Concretely, we apply T-CT to ImageNet-pretrained Swin-T and Swin-S models, both of which use GELU activations in their feed-forward blocks. As before, we transfer these models to the same 12 downstream datasets and compare T-CT against linear probing on the frozen backbone and LoRA ($r = 1$, $\alpha = 1$). Additional training details are provided in Appendix B.4.

Table 4: Mean accuracy (%) over three runs of ImageNet-pretrained Swin-T/S when transferred to 12 downstream datasets. The second row under each method indicates the number of trainable parameters (excluding the linear classifier). **T-CT improves over linear probing but underperforms LoRA.** Full results (± std) in Table S9.

| Dataset | Swin-T | | | Swin-S | | |
|---|---|---|---|---|---|---|
| | Frozen (0) | LoRA (74832) | T-CT (532) | Frozen (0) | LoRA (148560) | T-CT (868) |
| Arabic Characters | 83.48 | **93.24** | 85.02 | 83.83 | **94.38** | 86.65 |
| Arabic Digits | 98.14 | **99.19** | 98.47 | 98.28 | **99.19** | 98.39 |
| Beans | 88.28 | **94.01** | 89.06 | 90.89 | **95.05** | 91.41 |
| CUB-200 | 73.42 | **78.73** | 74.33 | 72.66 | **79.45** | 73.40 |
| DTD | 70.66 | 70.99 | **71.45** | 69.77 | 71.56 | **72.43** |
| FashionMNIST | 89.89 | **93.15** | 90.23 | 89.75 | **93.52** | 89.85 |
| FGVC-Aircraft | 48.06 | **48.29** | 47.58 | 44.36 | **51.94** | 45.72 |
| Flowers102 | 86.66 | **90.22** | 85.35 | 83.24 | **87.67** | 85.08 |
| Food101 | 77.05 | **83.69** | 78.90 | 77.59 | **85.17** | 79.45 |
| DermaMNIST | 75.83 | **76.71** | 75.86 | 76.64 | **78.15** | 77.14 |
| OCTMNIST | 69.97 | **76.30** | 67.97 | 66.90 | **76.97** | 69.07 |
| PathMNIST | 89.14 | **92.26** | 91.73 | 89.74 | **92.79** | 92.13 |
| *Average* | 79.22 | **83.06** | 79.66 | 78.64 | **83.82** | 80.06 |

The results, presented in Table 4, show that T-CT yields average relative improvements over linear probing of 0.48% and 1.94% on Swin-T and Swin-S, respectively. However, it underperforms LoRA in this setting, trailing by 4.01% and 4.76% on the two models. Nonetheless, as shown in Table 4, T-CT requires only 0.71% and 0.58% as many trainable parameters as LoRA on Swin-T and Swin-S—a much lower ratio than in the ResNet experiments. Thus, despite its lower performance, the results still highlight its potential for transformer architectures. It is also worth noting that in the current implementation, CT is applied only to the feed-forward blocks, which constitute a relatively small portion of the transformer, while the attention layers make up the majority. Extending CT to modulate the curvature within attention mechanisms—such as by tuning the temperature in the softmax of the attention block—is an interesting future work.

## 5 Conclusion

In this paper, we propose Curvature Tuning (CT), an interpretable and principled model steering method that provably modulates a network's decision boundary through a single parameter injected into its activation functions—without altering the model weights. Theoretically, we show that CT adjusts a model's nonlinearities and effectively projects it onto a space of smoother functions, offering a complementary perspective to existing PEFT methods. Practically, we introduce two variants: a steering form with fixed parameters (S-CT) and a finetuning form with learnable ones (T-CT). Both improve model generalization and exhibit an implicit bias toward enhancing model robustness.

While promising, CT also presents open questions for future work. In particular, exploring how to integrate the method more effectively with popular components such as softmax-based attention as well as kernelized attention remains an exciting direction for future work.

## Broader impacts

This paper presents work whose goal is to advance the field of deep learning. There are many potential societal consequences of our work, none of which we feel must be specifically highlighted here.

## Acknowledgments and disclosure of funding

Computations were in part enabled by the Berzelius resource provided by the Knut and Alice Wallenberg Foundation at the National Supercomputer Centre.

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

# Appendix

The remainder of the paper presents complementary background, experimental validation, and theoretical derivations that support our main results. The appendix is organized as follows.

1. Appendix A briefly connects several deep network architectures to affine spline operators.
2. Appendix B details our experimental setup and results.
3. Appendix C provides theoretical intuition behind CT.
4. Appendix D provides pseudocode for S-CT and T-CT.
5. Appendix E provides pseudocode for LoRA, describing how the method was applied throughout our experiments.

## A   Spline theory

The spline theory of deep learning establishes that a large class of deep network (DN) layers can be modeled as Max Affine Spline Operators (MASO). More precisely:

**Theorem A.1.** *(Propositions 1-4 in [13]) Any DN layer comprising a linear operator (e.g., fully connected or convolutional layer) followed by a convex and piecewise affine nonlinear operator (e.g., ReLU, leaky-ReLU, absolute value activation, max/average/channel pooling, maxout; with or without skip connections) is a MASO.*

Consequently, a deep network (e.g., MLP, CNN, RNN, ResNet) composed of such linear operators and convex, piecewise affine nonlinear operators is a composition of MASOs. However, it is important to note that the network as a whole is not a MASO but an Affine Spline Operator (ASO). In other words, conditioned on the input, such deep networks are equivalent to an affine operator, but globally, the induced mapping is not convex.

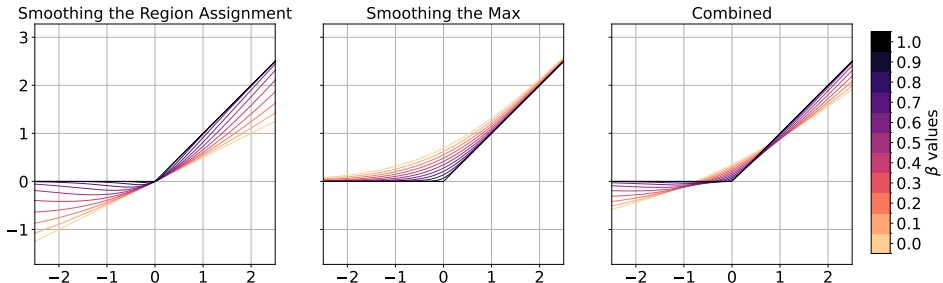

Figure S1: Visualization of nonlinearity smoothing through region assignment smoothing, max smoothing, and their combination. For a ReLU network, **the combined approach mitigates the opposing biases introduced by the individual methods.**

Building on the MASO interpretation, curvature tuning proposes to smoothen nonlinearities (e.g. ReLU) of a DN as a novel form of model steering that avoids retraining or finetuning the learned layers. By recalling Section 3.1, when smoothing is performed by applying Eq. (5) or Eq. (7) to a DN layer (interpreted as a MASO), the layer's output is statistically biased by either a negative or a positive factor, respectively. In order to counter the bias without retraining, a convex combination of the two equations is used, as shown in Fig. S1 for different values of $\beta$.

## B   Supplementary experimental details

This section provides additional experimental setup details and results, organized to correspond with the subsections in Section 4.

All experiments were conducted using 8 RTX 3090 GPUs and one L40 GPU, with runs performed under random seeds 42, 43, and 44.

### B.1 Improving generalization on downstream datasets with S-CT (Section 4.1)

For each of the 12 downstream datasets, we split the data into training, validation, and test sets. If a dataset does not include a validation set, we hold out 20% of the training data using stratified sampling. Otherwise, we use the original validation split provided.

All linear classifiers are trained for 20 epochs using the Adam optimizer with a learning rate of $10^{-3}$. We apply linear warm-up during the first epoch and decay the learning rate by a factor of 10 after epoch 10.

Additional results are provided as follows:

- Table S1: mean accuracy (± std) over three runs of ImageNet-pretrained ResNet-18/50/152 and VGG-11 when transferred to 12 downstream datasets, comparing linear probing with and without S-CT.
- Table S2: mean optimal $\beta$ values (± std) of S-CT across three runs.
- Table S3: ablation experiments on the choice of $c = 0.5$ for S-CT.
- Fig. S2: example validation accuracy vs. $\beta$ curves over three runs for S-CT.

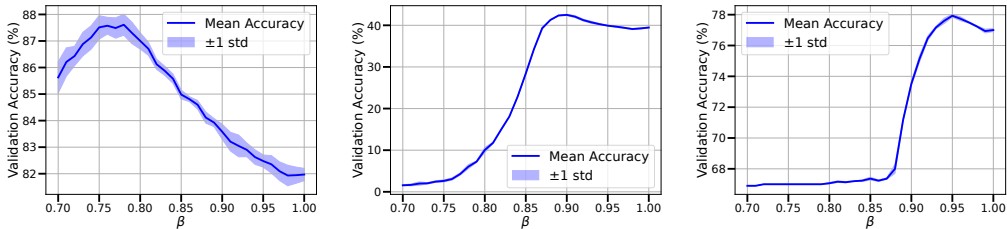

(a) ResNet-18 on Arabic Characters   (b) ResNet-50 on FGVC-Aircraft   (c) ResNet-152 on DermaMNIST

Figure S2: Validation accuracy (%) of S-CT during the $\beta$ search, averaged over three runs. **The accuracy curve varies smoothly and typically peaks in the middle of the $\beta$ range.**

### B.2 T-CT is comparable to LoRA with fewer parameters (Section 4.2)

Both T-CT and LoRA are trained for 20 epochs using the Adam optimizer. To ensure proper convergence, we use different learning rates: for T-CT, a learning rate of $10^{-1}$ is applied to the $(\beta, c)$ parameters and $10^{-3}$ to the linear classifier; for LoRA, a learning rate of $10^{-4}$ is used for both the adapter parameters and the classifier. As before, we apply linear warm-up during the first epoch and decay the learning rate by a factor of 10 after epoch 10.

Additional results are provided as follows:

- Table S1: mean accuracy (± std) over three runs of ImageNet-pretrained ResNet-18/50/152 and VGG-11 when transferred to 12 downstream datasets, comparing LoRA and T-CT.
- Table S4: additional experiments on LoRA with rank $r \in \{1, 2, 4\}$ and scaling factors $\alpha \in \{r, 2r, 4r\}$.
- Tables S5 and S6: mean (± std) of the learned $\beta$ and $c$ values of T-CT across three runs.
- Figs. S3 and S4: example distributions of $\beta$ and $c$ values in T-CT, illustrating commonly and uncommonly observed patterns.

### B.3 Improving model robustness through CT's implicit bias (Section 4.3)

Due to computational constraints, we evaluate each benchmark using 1,000 samples. For adversarial evaluations, we follow the official RobustBench settings: $\varepsilon_2 = 0.5$ for $\ell_2$ attacks and $\varepsilon_\infty = \frac{8}{255}$ for $\ell_\infty$ attacks.

Additional results are provided as follows:

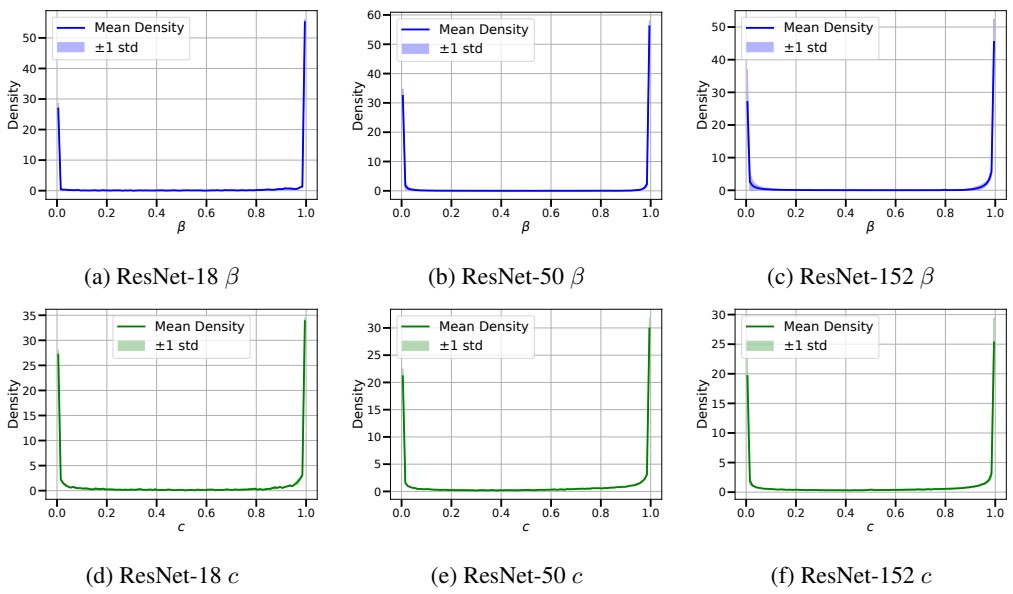

Figure S3: Common distributions of $\beta$ (top) and $c$ (bottom) in T-CT across ResNet-18/50/152, averaged over three runs (OCTMNIST shown as a representative dataset). **Both $\beta$ and $c$ consistently exhibit sharp U-shaped distributions that appear similar across all models.**

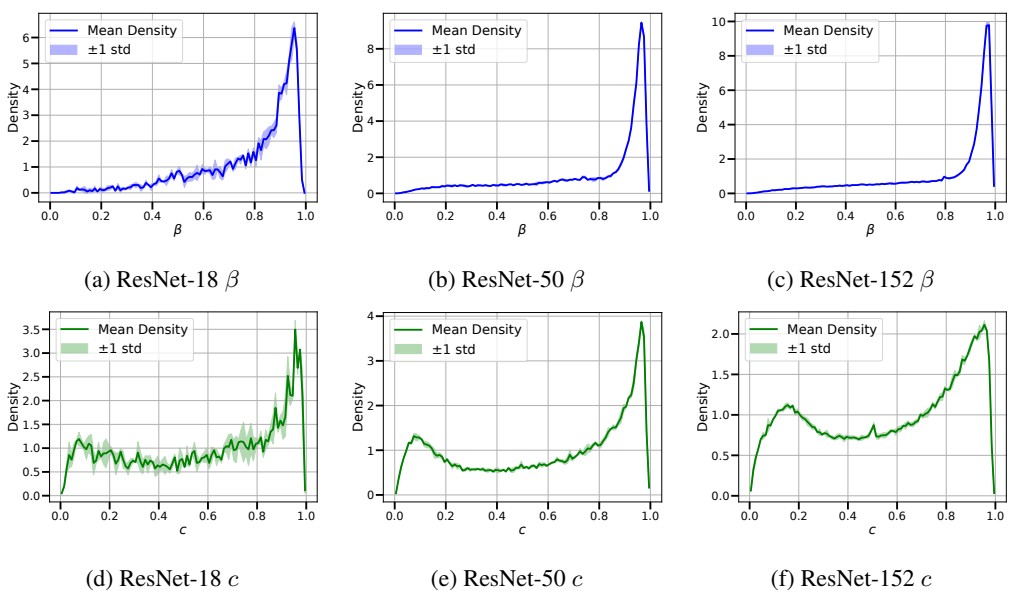

Figure S4: Uncommon distributions of $\beta$ (top) and $c$ (bottom) in T-CT across ResNet-18/50/152, averaged over three runs (DTD shown as an example dataset). **While the overall shape is dataset-specific, the distributions of both $\beta$ and $c$ remain consistent across models.**

- Table S7: mean robust accuracy (± std) over three runs of ImageNet-pretrained ResNet-18/50/152 under $\ell_\infty/\ell_2$ attacks and corruptions on CIFAR-10/100 and ImageNet.

- Table S8: mean robust accuracy (± std) over three runs of ImageNet-pretrained ResNet-18/50/152 transferred to CIFAR-10/100 under $\ell_\infty$, $\ell_2$ attacks, and corruptions.

## B.4   CT shows promise on transformers (Section 4.4)

In this experiment, LoRA is applied to all QKV projection layers. For all three methods—linear probing, LoRA, and T-CT—we perform a grid search over the learning rate and select the model achieving the best validation accuracy for testing. The learning rate is selected from $10^{-2}, 10^{-3}, 10^{-4}$ for linear probing, from $10^{-3}, 10^{-4}, 10^{-5}$ for LoRA, and for T-CT, we fix the learning rate of the linear classifier to $10^{-3}$ while searching over $10^{-1}, 10^{-2}, 10^{-3}, 10^{-4}$ for the $(\beta, c)$ parameters. All other training configurations follow those described in Appendices B.1 and B.2.

Additional results are provided as follows:

- Table S9: mean accuracy (± std) over three runs of ImageNet-pretrained Swin-T/S when transferred to 12 downstream datasets.

Table S1: Mean accuracy (%) ± standard deviation over three runs of ImageNet-pretrained ResNet-18/50/152 and VGG-11 when transferred to 12 downstream datasets. The second row under each method indicates the number of trainable parameters (excluding the linear classifier). **S-CT outperforms linear probing on the frozen backbone, and T-CT surpasses LoRA (rank 1).**

(a) ResNet-18

| Dataset | Frozen (0) | S-CT (1) | LoRA (35923) | T-CT (3968) |
|---|---|---|---|---|
| Arabic Characters | 81.91 ± 0.15 | 87.65 ± 0.18 | 93.37 ± 0.31 | **93.76 ± 0.22** |
| Arabic Digits | 97.93 ± 0.08 | 98.77 ± 0.01 | **99.08 ± 0.05** | 99.03 ± 0.01 |
| Beans | 87.76 ± 2.05 | 90.36 ± 1.19 | 93.23 ± 0.37 | **94.01 ± 0.37** |
| CUB-200 | 62.84 ± 0.29 | 63.18 ± 0.28 | 54.83 ± 0.37 | **64.30 ± 0.16** |
| DTD | 62.80 ± 0.42 | 62.66 ± 0.24 | 54.36 ± 0.31 | **63.62 ± 0.67** |
| FashionMNIST | 88.63 ± 0.13 | 88.70 ± 0.10 | **91.65 ± 0.12** | 91.07 ± 0.16 |
| FGVC-Aircraft | 36.80 ± 0.37 | 38.68 ± 0.05 | 29.19 ± 1.00 | **46.44 ± 0.49** |
| Flowers102 | 80.86 ± 0.29 | 81.97 ± 0.26 | 67.53 ± 0.76 | **86.55 ± 0.21** |
| Food101 | 61.41 ± 0.07 | 62.27 ± 0.25 | 64.40 ± 0.08 | **66.04 ± 0.17** |
| DermaMNIST | 74.83 ± 0.23 | 75.05 ± 0.60 | 74.21 ± 0.50 | **77.66 ± 0.29** |
| OCTMNIST | 65.03 ± 0.69 | 67.27 ± 0.23 | **74.27 ± 0.49** | 69.53 ± 1.11 |
| PathMNIST | 86.77 ± 0.04 | 87.51 ± 0.05 | **87.62 ± 0.12** | 87.17 ± 0.66 |
| *Average* | 73.96 | 75.34 | 73.64 | **78.26** |

(b) ResNet-50

| Dataset | Frozen (0) | S-CT (1) | LoRA (79443) | T-CT (45440) |
|---|---|---|---|---|
| Arabic Characters | 80.65 ± 0.07 | 83.66 ± 0.41 | 94.21 ± 0.28 | **95.67 ± 0.03** |
| Arabic Digits | 98.33 ± 0.02 | 98.37 ± 0.06 | 99.08 ± 0.00 | **99.16 ± 0.03** |
| Beans | 89.58 ± 0.74 | 91.93 ± 0.90 | 94.79 ± 0.74 | **95.57 ± 0.74** |
| CUB-200 | 65.23 ± 0.43 | 64.62 ± 0.32 | 66.17 ± 0.51 | **71.03 ± 0.64** |
| DTD | **67.34 ± 0.16** | 66.91 ± 0.14 | 64.70 ± 0.42 | 65.07 ± 0.37 |
| FashionMNIST | 90.05 ± 0.07 | 90.34 ± 0.23 | 92.19 ± 0.17 | **92.78 ± 0.06** |
| FGVC-Aircraft | 38.03 ± 0.32 | 41.16 ± 0.32 | 41.99 ± 0.03 | **55.70 ± 0.76** |
| Flowers102 | 84.00 ± 0.06 | 83.84 ± 0.13 | 82.58 ± 0.47 | **87.62 ± 0.28** |
| Food101 | 68.06 ± 0.11 | 68.02 ± 0.11 | 71.42 ± 0.14 | **73.60 ± 0.13** |
| DermaMNIST | 75.94 ± 0.12 | 75.89 ± 0.03 | 75.73 ± 0.72 | **78.02 ± 0.50** |
| OCTMNIST | 67.53 ± 0.21 | 68.00 ± 0.17 | **75.90 ± 0.33** | 74.13 ± 1.65 |
| PathMNIST | 90.08 ± 0.22 | **90.26 ± 0.20** | 85.43 ± 1.99 | 87.33 ± 0.74 |
| *Average* | 76.24 | 76.92 | 78.68 | **81.31** |

(c) ResNet-152

| Dataset | Frozen (0) | S-CT (1) | LoRA (243283) | T-CT (143744) |
|---|---|---|---|---|
| Arabic Characters | 79.86 ± 0.12 | 79.21 ± 0.55 | 95.96 ± 0.21 | **96.47 ± 0.39** |
| Arabic Digits | 98.07 ± 0.05 | 98.15 ± 0.10 | **99.15 ± 0.04** | 99.10 ± 0.05 |
| Beans | 87.50 ± 1.10 | 87.50 ± 0.78 | 93.75 ± 1.91 | **96.35 ± 1.33** |
| CUB-200 | 67.68 ± 0.54 | 68.15 ± 0.62 | 70.59 ± 0.72 | **73.04 ± 0.19** |
| DTD | 66.95 ± 0.03 | **66.97 ± 0.05** | 66.63 ± 0.07 | 63.39 ± 0.34 |
| FashionMNIST | 90.37 ± 0.11 | 90.44 ± 0.16 | 92.77 ± 0.04 | **93.39 ± 0.12** |
| FGVC-Aircraft | 38.74 ± 0.16 | 38.51 ± 0.14 | 48.84 ± 0.54 | **58.16 ± 0.31** |
| Flowers102 | 82.98 ± 0.16 | 83.28 ± 0.25 | **84.40 ± 0.74** | 83.43 ± 1.01 |
| Food101 | 71.11 ± 0.09 | 71.13 ± 0.08 | 74.66 ± 0.08 | **76.08 ± 0.15** |
| DermaMNIST | 75.68 ± 0.47 | 76.23 ± 0.14 | 76.91 ± 0.79 | **77.94 ± 0.60** |
| OCTMNIST | 69.27 ± 0.98 | 69.10 ± 1.47 | **76.43 ± 0.54** | 75.17 ± 2.10 |
| PathMNIST | **89.91 ± 0.12** | 89.82 ± 0.09 | 84.94 ± 1.27 | 83.60 ± 0.42 |
| *Average* | 76.51 | 76.54 | 80.42 | **81.34** |

(d) VGG-11

| Dataset | Frozen (0) | S-CT (1) | LoRA (60315) | T-CT (21888) |
|---|---|---|---|---|
| Arabic Characters | 81.19 ± 0.33 | 83.04 ± 0.35 | 88.88 ± 0.37 | **93.04 ± 0.50** |
| Arabic Digits | 97.97 ± 0.03 | 98.19 ± 0.01 | 98.76 ± 0.05 | **99.01 ± 0.04** |
| Beans | 91.67 ± 0.97 | 90.89 ± 0.37 | **93.75 ± 1.69** | 92.45 ± 1.95 |
| CUB-200 | 61.04 ± 0.29 | 60.97 ± 0.68 | 59.22 ± 0.14 | **63.13 ± 0.32** |
| DTD | 65.07 ± 0.45 | 65.05 ± 0.16 | 63.46 ± 0.76 | **65.25 ± 0.56** |
| FashionMNIST | 89.44 ± 0.11 | 89.35 ± 0.19 | **90.52 ± 0.15** | 90.49 ± 0.16 |
| FGVC-Aircraft | 39.48 ± 0.09 | 41.34 ± 0.40 | 39.29 ± 1.02 | **47.79 ± 0.23** |
| Flowers102 | 80.32 ± 0.21 | 80.44 ± 0.30 | 78.38 ± 0.59 | **84.31 ± 0.20** |
| Food101 | 61.03 ± 0.22 | 60.94 ± 0.03 | 64.22 ± 0.43 | **66.43 ± 0.05** |
| DermaMNIST | 75.94 ± 0.29 | 76.24 ± 0.06 | 74.06 ± 0.22 | **77.97 ± 0.38** |
| OCTMNIST | 67.33 ± 0.92 | 68.13 ± 0.12 | **72.13 ± 1.18** | 70.77 ± 0.45 |
| PathMNIST | 86.85 ± 0.14 | 87.55 ± 0.03 | **89.23 ± 0.43** | 88.90 ± 0.53 |
| *Average* | 74.78 | 75.18 | 75.99 | **78.30** |

Table S2: Mean $\beta$ ± standard deviation of S-CT over three runs of ImageNet-pretrained ResNet-18/50/152 and VGG-11 when transferred to 12 downstream datasets. **The average optimal $\beta$ values are consistently high, ranging from 0.84 to 0.96 across models.**

| Dataset | ResNet-18 | ResNet-50 | ResNet-152 | VGG-11 |
|---|---|---|---|---|
| Arabic Characters | 0.77 ± 0.01 | 0.89 ± 0.01 | 0.96 ± 0.00 | 0.88 ± 0.02 |
| Arabic Digits | 0.75 ± 0.01 | 0.93 ± 0.04 | 0.95 ± 0.01 | 0.85 ± 0.01 |
| Beans | 0.76 ± 0.02 | 0.94 ± 0.01 | 0.98 ± 0.02 | 0.76 ± 0.00 |
| CUB-200 | 0.91 ± 0.02 | 0.93 ± 0.01 | 0.94 ± 0.01 | 0.86 ± 0.02 |
| DTD | 0.88 ± 0.02 | 0.98 ± 0.01 | 1.00 ± 0.00 | 0.93 ± 0.01 |
| FashionMNIST | 0.92 ± 0.01 | 0.95 ± 0.00 | 0.97 ± 0.02 | 0.98 ± 0.02 |
| FGVC-Aircraft | 0.82 ± 0.02 | 0.90 ± 0.00 | 0.98 ± 0.03 | 0.82 ± 0.04 |
| Flowers102 | 0.84 ± 0.02 | 0.96 ± 0.01 | 0.95 ± 0.00 | 0.92 ± 0.02 |
| Food101 | 0.87 ± 0.01 | 0.98 ± 0.00 | 0.99 ± 0.01 | 0.97 ± 0.02 |
| DermaMNIST | 0.94 ± 0.08 | 0.95 ± 0.00 | 0.95 ± 0.00 | 0.93 ± 0.02 |
| OCTMNIST | 0.80 ± 0.00 | 0.94 ± 0.01 | 0.99 ± 0.01 | 0.96 ± 0.00 |
| PathMNIST | 0.83 ± 0.00 | 0.98 ± 0.01 | 0.92 ± 0.00 | 0.91 ± 0.00 |
| *Average* | 0.84 | 0.94 | 0.96 | 0.90 |

Table S3: Mean accuracy (%) ± standard deviation over three runs of ImageNet-pretrained ResNet-18/50/152 and VGG-11 when transferred to 12 downstream datasets. The second row under each method indicates the number of trainable parameters (excluding the linear classifier). **By setting $c = 0.5$, S-CT performs better than SiLU ($c = 1$) and Softplus ($c = 0$) on ResNet-18 and ResNet-50, but slightly worse on ResNet-152.**

(a) ResNet-18

| Dataset | Frozen (0) | S-CT (1) | SiLU (1) | Softplus (1) |
|---|---|---|---|---|
| Arabic Characters | 81.91 ± 0.15 | **87.65 ± 0.18** | 81.91 ± 0.19 | 84.89 ± 0.12 |
| Arabic Digits | 97.93 ± 0.08 | **98.77 ± 0.01** | 97.95 ± 0.11 | 98.64 ± 0.06 |
| Beans | 87.76 ± 2.05 | **90.36 ± 1.19** | 88.80 ± 1.19 | 88.02 ± 1.19 |
| CUB-200 | 62.84 ± 0.29 | **63.18 ± 0.28** | 62.90 ± 0.33 | 63.08 ± 0.11 |
| DTD | 62.80 ± 0.42 | 62.66 ± 0.24 | 62.89 ± 0.33 | **63.24 ± 0.23** |
| FashionMNIST | 88.63 ± 0.13 | 88.70 ± 0.10 | 88.63 ± 0.16 | **88.93 ± 0.09** |
| FGVC-Aircraft | 36.80 ± 0.37 | **38.68 ± 0.05** | 36.29 ± 0.24 | 37.44 ± 0.68 |
| Flowers102 | 80.86 ± 0.29 | 81.97 ± 0.26 | 80.86 ± 0.35 | **83.14 ± 0.03** |
| Food101 | 61.41 ± 0.07 | 62.27 ± 0.25 | 61.37 ± 0.05 | **62.76 ± 0.06** |
| DermaMNIST | 74.83 ± 0.23 | 75.05 ± 0.60 | 74.63 ± 0.19 | **75.13 ± 0.22** |
| OCTMNIST | 65.03 ± 0.69 | **67.27 ± 0.23** | 65.10 ± 0.82 | 66.57 ± 0.75 |
| PathMNIST | 86.77 ± 0.04 | 87.51 ± 0.05 | 86.77 ± 0.05 | **87.84 ± 0.25** |
| *Average* | 73.96 | **75.34** | 74.01 | 74.97 |

(b) ResNet-50

| Dataset | Frozen (0) | S-CT (1) | SiLU (1) | Softplus (1) |
|---|---|---|---|---|
| Arabic Characters | 80.65 ± 0.07 | 83.66 ± 0.41 | **86.46 ± 0.03** | 81.10 ± 0.21 |
| Arabic Digits | 98.33 ± 0.02 | 98.37 ± 0.06 | **98.56 ± 0.11** | 98.41 ± 0.15 |
| Beans | 89.58 ± 0.74 | **91.93 ± 0.90** | 86.46 ± 1.19 | 91.15 ± 0.90 |
| CUB-200 | 65.23 ± 0.43 | 64.62 ± 0.32 | 65.12 ± 0.53 | **65.61 ± 0.50** |
| DTD | 67.34 ± 0.16 | 66.91 ± 0.14 | **67.27 ± 0.08** | 67.06 ± 0.62 |
| FashionMNIST | 90.05 ± 0.07 | 90.34 ± 0.23 | 89.96 ± 0.01 | **90.35 ± 0.11** |
| FGVC-Aircraft | 38.03 ± 0.32 | **41.16 ± 0.32** | 38.03 ± 0.39 | 40.09 ± 0.02 |
| Flowers102 | 84.00 ± 0.06 | 83.84 ± 0.13 | **84.09 ± 0.11** | 83.94 ± 0.14 |
| Food101 | 68.06 ± 0.11 | 68.02 ± 0.11 | **68.14 ± 0.06** | 67.90 ± 0.15 |
| DermaMNIST | 75.94 ± 0.12 | **75.89 ± 0.03** | 75.79 ± 0.12 | 75.79 ± 0.32 |
| OCTMNIST | 67.53 ± 0.21 | **68.00 ± 0.17** | 67.33 ± 0.15 | 67.00 ± 0.20 |
| PathMNIST | 90.08 ± 0.22 | **90.26 ± 0.20** | 89.95 ± 0.13 | 89.99 ± 0.14 |
| *Average* | 76.24 | **76.92** | 76.43 | 76.53 |

(c) ResNet-152

| Dataset | Frozen (0) | S-CT (1) | SiLU (1) | Softplus (1) |
|---|---|---|---|---|
| Arabic Characters | 79.86 ± 0.12 | 79.21 ± 0.55 | **80.24 ± 0.44** | 79.83 ± 0.27 |
| Arabic Digits | 98.07 ± 0.05 | 98.15 ± 0.10 | **98.20 ± 0.18** | 97.98 ± 0.01 |
| Beans | 87.50 ± 1.10 | 87.50 ± 0.78 | **88.80 ± 1.97** | 88.80 ± 2.39 |
| CUB-200 | 67.68 ± 0.54 | 68.15 ± 0.62 | 67.68 ± 0.66 | **69.80 ± 0.32** |
| DTD | 66.95 ± 0.03 | 66.97 ± 0.05 | 66.86 ± 0.23 | 66.95 ± 0.03 |
| FashionMNIST | 90.37 ± 0.11 | 90.44 ± 0.16 | 90.28 ± 0.26 | **90.60 ± 0.03** |
| FGVC-Aircraft | 38.74 ± 0.16 | 38.51 ± 0.14 | 38.74 ± 0.20 | **39.20 ± 0.38** |
| Flowers102 | 82.98 ± 0.16 | 83.28 ± 0.25 | 82.97 ± 0.19 | **83.54 ± 0.13** |
| Food101 | 71.11 ± 0.09 | 71.13 ± 0.08 | 71.11 ± 0.11 | **71.19 ± 0.08** |
| DermaMNIST | 75.68 ± 0.47 | **76.23 ± 0.14** | 74.76 ± 0.69 | 76.16 ± 0.10 |
| OCTMNIST | 69.27 ± 0.98 | 69.10 ± 1.47 | 69.47 ± 1.29 | **69.90 ± 0.50** |
| PathMNIST | 89.91 ± 0.12 | 89.82 ± 0.09 | 89.91 ± 0.15 | **90.75 ± 0.06** |
| *Average* | 76.51 | 76.54 | 76.59 | **77.06** |

Table S4: Mean accuracy (%) ± standard deviation over three runs of ImageNet-pretrained ResNet-18/50/152 when transferred to 12 downstream datasets. The second row under each method indicates the number of trainable parameters (excluding the linear classifier). For each dataset and LoRA rank $r$, we report the best test accuracy achieved among the candidate scaling factors $\alpha \in \{r, 2r, 4r\}$. **T-CT can still outperform LoRA under this more challenging setting.**

(a) ResNet-18

| Dataset | T-CT (3968) | LoRA ($r = 1$) (35923) | LoRA ($r = 2$) (71846) | LoRA ($r = 4$) (143692) |
|---|---|---|---|---|
| Arabic Characters | 93.76 ± 0.22 | 94.23 ± 0.13 | 95.30 ± 0.18 | **96.26 ± 0.12** |
| Arabic Digits | 99.03 ± 0.01 | 99.12 ± 0.02 | 99.21 ± 0.05 | **99.23 ± 0.03** |
| Beans | 94.01 ± 0.37 | 94.01 ± 0.74 | **95.83 ± 0.37** | **95.83 ± 1.33** |
| CUB-200 | **64.30 ± 0.16** | 54.83 ± 0.37 | 56.11 ± 0.26 | 57.47 ± 0.51 |
| DTD | **63.62 ± 0.67** | 54.36 ± 0.31 | 56.17 ± 0.22 | 57.93 ± 0.43 |
| FashionMNIST | 91.07 ± 0.16 | 92.03 ± 0.11 | 92.83 ± 0.04 | **93.50 ± 0.05** |
| FGVC-Aircraft | **46.44 ± 0.49** | 29.50 ± 0.92 | 32.94 ± 0.40 | 39.13 ± 0.32 |
| Flowers102 | **86.55 ± 0.21** | 67.53 ± 0.76 | 69.68 ± 0.89 | 73.30 ± 0.46 |
| Food101 | 66.04 ± 0.17 | 64.40 ± 0.08 | 65.96 ± 0.32 | **66.97 ± 0.35** |
| DermaMNIST | **77.66 ± 0.29** | 75.54 ± 0.20 | 76.79 ± 0.77 | 76.72 ± 0.51 |
| OCTMNIST | 69.53 ± 1.11 | 74.83 ± 0.17 | 76.37 ± 0.66 | **76.47 ± 0.26** |
| PathMNIST | 87.17 ± 0.66 | 87.78 ± 0.18 | 88.08 ± 1.15 | **88.85 ± 0.48** |
| *Average* | **78.26** | 74.01 | 75.44 | 76.80 |

(b) ResNet-50

| Dataset | T-CT (45440) | LoRA ($r = 1$) (79443) | LoRA ($r = 2$) (158886) | LoRA ($r = 4$) (317772) |
|---|---|---|---|---|
| Arabic Characters | 95.67 ± 0.03 | 94.38 ± 0.22 | 95.67 ± 0.34 | **96.30 ± 0.04** |
| Arabic Digits | 99.16 ± 0.03 | 99.09 ± 0.01 | **99.22 ± 0.09** | **99.22 ± 0.02** |
| Beans | 95.57 ± 0.74 | **96.35 ± 0.37** | 96.09 ± 1.69 | 95.31 ± 0.64 |
| CUB-200 | **71.03 ± 0.64** | 66.17 ± 0.51 | 67.91 ± 0.53 | 68.93 ± 0.23 |
| DTD | 65.07 ± 0.37 | 64.79 ± 0.30 | 64.91 ± 0.49 | **67.07 ± 0.31** |
| FashionMNIST | 92.78 ± 0.06 | 92.19 ± 0.17 | 92.90 ± 0.14 | **93.59 ± 0.13** |
| FGVC-Aircraft | **55.70 ± 0.76** | 42.12 ± 0.17 | 47.46 ± 0.19 | 52.64 ± 0.47 |
| Flowers102 | **87.62 ± 0.28** | 82.58 ± 0.47 | 83.39 ± 0.35 | 84.63 ± 0.29 |
| Food101 | 73.60 ± 0.13 | 71.42 ± 0.14 | 73.01 ± 0.24 | **74.89 ± 0.05** |
| DermaMNIST | **78.02 ± 0.50** | 76.21 ± 0.27 | 77.26 ± 0.36 | 77.39 ± 0.47 |
| OCTMNIST | 74.13 ± 1.65 | 76.23 ± 0.09 | 76.07 ± 1.19 | **77.83 ± 0.82** |
| PathMNIST | 87.33 ± 0.74 | 87.29 ± 0.67 | 86.04 ± 0.23 | **87.44 ± 0.16** |
| *Average* | **81.31** | 79.07 | 79.99 | 81.27 |

(c) ResNet-152

| Dataset | T-CT (143744) | LoRA ($r = 1$) (243283) | LoRA ($r = 2$) (486566) | LoRA ($r = 4$) (973132) |
|---|---|---|---|---|
| Arabic Characters | 96.47 ± 0.39 | 96.00 ± 0.16 | 96.43 ± 0.02 | **96.87 ± 0.11** |
| Arabic Digits | 99.10 ± 0.05 | 99.16 ± 0.03 | 99.21 ± 0.03 | **99.25 ± 0.01** |
| Beans | 96.35 ± 1.33 | 94.53 ± 1.10 | **97.92 ± 0.37** | 97.14 ± 0.74 |
| CUB-200 | **73.04 ± 0.19** | 70.75 ± 0.59 | 70.94 ± 0.15 | 71.72 ± 0.43 |
| DTD | 63.39 ± 0.34 | 66.63 ± 0.07 | 67.66 ± 0.50 | **68.28 ± 0.51** |
| FashionMNIST | 93.39 ± 0.12 | 92.77 ± 0.04 | 93.49 ± 0.04 | **93.98 ± 0.14** |
| FGVC-Aircraft | 58.16 ± 0.31 | 49.06 ± 0.26 | 55.82 ± 1.04 | **59.98 ± 0.26** |
| Flowers102 | 83.43 ± 1.01 | 84.51 ± 0.58 | 84.83 ± 0.17 | **86.24 ± 0.04** |
| Food101 | 76.08 ± 0.15 | 74.66 ± 0.08 | 76.00 ± 0.16 | **76.86 ± 0.10** |
| DermaMNIST | **77.94 ± 0.60** | 77.02 ± 0.69 | 77.31 ± 0.75 | 77.46 ± 0.16 |
| OCTMNIST | 75.17 ± 2.10 | 77.90 ± 0.36 | 78.23 ± 1.32 | **78.63 ± 0.21** |
| PathMNIST | 83.60 ± 0.42 | 86.75 ± 0.86 | **88.33 ± 0.33** | 86.81 ± 1.95 |
| *Average* | 81.34 | 80.81 | 82.18 | **82.77** |

Table S5: Distribution of $\beta$ values in T-CT, computed over all $\beta$ parameters across all three runs of ImageNet-pretrained ResNet-18/50/152 and VGG-11 when transferred to 12 downstream datasets. **The mean and standard deviation of $\beta$ are similar across models (means between 0.69–0.79, stds between 0.31–0.37), suggesting consistent tuning behavior at the model level, while the relatively large standard deviations indicate substantial variation of $\beta$ within each network.**

| Dataset | ResNet-18 | ResNet-50 | ResNet-152 | VGG-11 |
|---|---|---|---|---|
| Arabic Characters | 0.72 ± 0.34 | 0.65 ± 0.41 | 0.68 ± 0.39 | 0.73 ± 0.39 |
| Arabic Digits | 0.70 ± 0.43 | 0.62 ± 0.48 | 0.62 ± 0.47 | 0.73 ± 0.43 |
| Beans | 0.72 ± 0.26 | 0.76 ± 0.23 | 0.77 ± 0.19 | 0.79 ± 0.20 |
| CUB-200 | 0.81 ± 0.17 | 0.76 ± 0.29 | 0.79 ± 0.29 | 0.75 ± 0.31 |
| DTD | 0.78 ± 0.19 | 0.77 ± 0.25 | 0.79 ± 0.24 | 0.80 ± 0.24 |
| FashionMNIST | 0.72 ± 0.41 | 0.65 ± 0.46 | 0.63 ± 0.46 | 0.81 ± 0.37 |
| FGVC-Aircraft | 0.75 ± 0.23 | 0.70 ± 0.33 | 0.74 ± 0.32 | 0.74 ± 0.31 |
| Flowers102 | 0.75 ± 0.16 | 0.75 ± 0.21 | 0.79 ± 0.17 | 0.74 ± 0.22 |
| Food101 | 0.80 ± 0.30 | 0.71 ± 0.43 | 0.76 ± 0.40 | 0.88 ± 0.27 |
| DermaMNIST | 0.74 ± 0.34 | 0.70 ± 0.39 | 0.70 ± 0.37 | 0.81 ± 0.33 |
| OCTMNIST | 0.67 ± 0.45 | 0.62 ± 0.48 | 0.63 ± 0.47 | 0.83 ± 0.35 |
| PathMNIST | 0.69 ± 0.43 | 0.65 ± 0.47 | 0.61 ± 0.48 | 0.82 ± 0.37 |
| *Average* | 0.74 ± 0.31 | 0.69 ± 0.37 | 0.71 ± 0.35 | 0.79 ± 0.32 |

Table S6: Distribution of $c$ values in T-CT, computed over all $c$ parameters across all three runs of ImageNet-pretrained ResNet-18/50/152 and VGG-11 when transferred to 12 downstream datasets. **The mean and standard deviation of $c$ are similar across models (means between 0.57–0.61, stds between 0.32–0.38), suggesting consistent tuning behavior at the model level, while the relatively large standard deviations indicate substantial variation of $c$ within each network.**

| Dataset | ResNet-18 | ResNet-50 | ResNet-152 | VGG-11 |
|---|---|---|---|---|
| Arabic Characters | 0.63 ± 0.39 | 0.61 ± 0.39 | 0.57 ± 0.37 | 0.65 ± 0.33 |
| Arabic Digits | 0.59 ± 0.43 | 0.57 ± 0.42 | 0.55 ± 0.41 | 0.62 ± 0.38 |
| Beans | 0.61 ± 0.29 | 0.54 ± 0.25 | 0.53 ± 0.23 | 0.55 ± 0.21 |
| CUB-200 | 0.60 ± 0.37 | 0.63 ± 0.37 | 0.60 ± 0.34 | 0.67 ± 0.31 |
| DTD | 0.59 ± 0.31 | 0.60 ± 0.32 | 0.57 ± 0.30 | 0.59 ± 0.27 |
| FashionMNIST | 0.55 ± 0.44 | 0.60 ± 0.42 | 0.56 ± 0.42 | 0.61 ± 0.38 |
| FGVC-Aircraft | 0.61 ± 0.36 | 0.63 ± 0.37 | 0.58 ± 0.35 | 0.66 ± 0.31 |
| Flowers102 | 0.58 ± 0.26 | 0.54 ± 0.26 | 0.54 ± 0.23 | 0.60 ± 0.22 |
| Food101 | 0.46 ± 0.47 | 0.63 ± 0.44 | 0.60 ± 0.43 | 0.58 ± 0.40 |
| DermaMNIST | 0.58 ± 0.38 | 0.59 ± 0.37 | 0.57 ± 0.36 | 0.56 ± 0.30 |
| OCTMNIST | 0.55 ± 0.45 | 0.60 ± 0.42 | 0.57 ± 0.42 | 0.62 ± 0.39 |
| PathMNIST | 0.51 ± 0.45 | 0.58 ± 0.43 | 0.57 ± 0.42 | 0.58 ± 0.40 |
| *Average* | 0.57 ± 0.38 | 0.59 ± 0.37 | 0.57 ± 0.36 | 0.61 ± 0.32 |

Table S7: Mean robust accuracy (%) ± standard deviation over three runs of ImageNet-pretrained ResNet-18/50/152 under $\ell_2/\ell_\infty$ attacks and corruptions on CIFAR-10/100 and ImageNet. **S-CT yields substantial improvements under $\ell_\infty$ attacks, with the selected $\beta$ values close to 1.**

| Robustness | Model | Dataset | Frozen | S-CT | $\beta$ |
|---|---|---|---|---|---|
| $\ell_2$ | ResNet-18 | CIFAR-10 | 53.67 ± 0.32 | 53.67 ± 0.32 | 1.00 ± 0.00 |
| | | CIFAR-100 | 24.30 ± 0.10 | **25.50 ± 0.00** | 0.92 ± 0.00 |
| | | ImageNet | 23.37 ± 0.06 | 23.37 ± 0.06 | 1.00 ± 0.00 |
| | | *Average* | 33.78 | **34.18** | 0.97 |
| | ResNet-50 | CIFAR-10 | 55.10 ± 0.10 | **56.53 ± 0.21** | 0.97 ± 0.00 |
| | | CIFAR-100 | 23.83 ± 0.06 | **25.80 ± 0.20** | 0.96 ± 0.00 |
| | | ImageNet | 31.90 ± 0.00 | 31.90 ± 0.00 | 1.00 ± 0.00 |
| | | *Average* | 36.94 | **38.08** | 0.98 |
| | ResNet-152 | CIFAR-10 | 56.27 ± 0.23 | 56.27 ± 0.23 | 1.00 ± 0.00 |
| | | CIFAR-100 | 27.90 ± 0.10 | **28.23 ± 0.12** | 0.98 ± 0.00 |
| | | ImageNet | 42.50 ± 0.00 | 42.50 ± 0.00 | 1.00 ± 0.00 |
| | | *Average* | 42.22 | **42.33** | 0.99 |
| $\ell_\infty$ | ResNet-18 | CIFAR-10 | 11.17 ± 0.06 | **14.93 ± 0.06** | 0.90 ± 0.00 |
| | | CIFAR-100 | 4.47 ± 0.06 | **6.90 ± 0.00** | 0.92 ± 0.00 |
| | | ImageNet | 0.00 ± 0.00 | **7.00 ± 0.10** | 0.89 ± 0.00 |
| | | *Average* | 5.21 | **9.61** | 0.90 |
| | ResNet-50 | CIFAR-10 | 10.10 ± 0.17 | **14.83 ± 0.06** | 0.95 ± 0.00 |
| | | CIFAR-100 | 4.43 ± 0.06 | **7.90 ± 0.00** | 0.93 ± 0.00 |
| | | ImageNet | 0.30 ± 0.00 | **9.30 ± 0.17** | 0.93 ± 0.00 |
| | | *Average* | 4.94 | **9.76** | 0.94 |
| | ResNet-152 | CIFAR-10 | 11.47 ± 0.06 | **15.00 ± 0.20** | 0.99 ± 0.00 |
| | | CIFAR-100 | 5.40 ± 0.00 | **7.70 ± 0.17** | 0.99 ± 0.00 |
| | | ImageNet | 0.30 ± 0.00 | **13.53 ± 0.06** | 0.97 ± 0.01 |
| | | *Average* | 5.72 | **12.08** | 0.98 |
| Corruptions | ResNet-18 | CIFAR-10 | 77.73 ± 0.00 | 77.73 ± 0.00 | 1.00 ± 0.00 |
| | | CIFAR-100 | 51.81 ± 0.00 | **51.95 ± 0.00** | 0.94 ± 0.00 |
| | | ImageNet | 33.11 ± 0.00 | **33.32 ± 0.00** | 0.92 ± 0.00 |
| | | *Average* | 54.22 | **54.33** | 0.95 |
| | ResNet-50 | CIFAR-10 | 77.26 ± 0.00 | 77.26 ± 0.00 | 1.00 ± 0.00 |
| | | CIFAR-100 | 53.91 ± 0.00 | **53.93 ± 0.00** | 0.98 ± 0.00 |
| | | ImageNet | 39.64 ± 0.00 | 39.64 ± 0.00 | 1.00 ± 0.00 |
| | | *Average* | 56.94 | 56.94 | 0.99 |
| | ResNet-152 | CIFAR-10 | 78.82 ± 0.00 | **78.83 ± 0.00** | 0.99 ± 0.00 |
| | | CIFAR-100 | 56.12 ± 0.00 | 56.12 ± 0.00 | 1.00 ± 0.00 |
| | | ImageNet | 45.47 ± 0.00 | 45.47 ± 0.00 | 0.99 ± 0.00 |
| | | *Average* | 60.14 | 60.14 | 0.99 |

Table S8: Mean robust accuracy (%) ± standard deviation over three runs of ImageNet-pretrained ResNet-18/50/152 transferred to CIFAR-10/100 under $\ell_2$, $\ell_\infty$ attacks, and corruptions. **T-CT improves $\ell_\infty$ robustness significantly compared to linear probing and LoRA.**

| Robustness | Model | Dataset | Frozen | LoRA | T-CT |
|---|---|---|---|---|---|
| $\ell_2$ | ResNet18 | CIFAR10 | 8.47 ± 0.26 | 5.93 ± 1.65 | **8.93 ± 0.37** |
| | | CIFAR100 | **1.57 ± 0.21** | 0.77 ± 0.33 | 1.10 ± 0.45 |
| | | *Average* | **5.02** | 3.35 | 5.01 |
| | ResNet50 | CIFAR10 | 6.23 ± 0.34 | 4.57 ± 1.32 | **6.83 ± 1.48** |
| | | CIFAR100 | **0.70 ± 0.08** | 0.37 ± 0.26 | 0.47 ± 0.31 |
| | | *Average* | 3.47 | 2.47 | **3.65** |
| | ResNet152 | CIFAR10 | **8.03 ± 0.52** | 4.63 ± 2.01 | 8.00 ± 1.22 |
| | | CIFAR100 | **0.90 ± 0.08** | 0.47 ± 0.26 | 0.50 ± 0.08 |
| | | *Average* | **4.46** | 2.55 | 4.25 |
| $\ell_\infty$ | ResNet18 | CIFAR10 | 0.30 ± 0.00 | 0.70 ± 0.71 | **1.57 ± 0.74** |
| | | CIFAR100 | 0.03 ± 0.05 | 0.07 ± 0.05 | **0.17 ± 0.12** |
| | | *Average* | 0.16 | 0.38 | **0.87** |
| | ResNet50 | CIFAR10 | 0.20 ± 0.08 | 0.33 ± 0.29 | **2.43 ± 1.54** |
| | | CIFAR100 | 0.00 ± 0.00 | 0.03 ± 0.05 | **0.07 ± 0.09** |
| | | *Average* | 0.10 | 0.18 | **1.25** |
| | ResNet152 | CIFAR10 | 0.43 ± 0.09 | 0.20 ± 0.14 | **5.10 ± 2.97** |
| | | CIFAR100 | **0.17 ± 0.05** | 0.00 ± 0.00 | 0.00 ± 0.00 |
| | | *Average* | 0.30 | 0.10 | **2.55** |
| Corruptions | ResNet18 | CIFAR10 | **21.34 ± 0.29** | 13.59 ± 0.30 | 16.83 ± 2.36 |
| | | CIFAR100 | **5.10 ± 0.15** | 2.96 ± 1.05 | 4.62 ± 0.68 |
| | | *Average* | **13.22** | 8.28 | 10.72 |
| | ResNet50 | CIFAR10 | **16.23 ± 0.21** | 11.69 ± 0.90 | 12.68 ± 2.06 |
| | | CIFAR100 | **3.47 ± 0.09** | 2.04 ± 0.36 | 1.61 ± 0.13 |
| | | *Average* | **9.85** | 6.86 | 7.14 |
| | ResNet152 | CIFAR10 | **13.82 ± 0.49** | 11.33 ± 1.22 | 9.83 ± 2.07 |
| | | CIFAR100 | 2.07 ± 0.12 | **2.13 ± 0.22** | 1.72 ± 0.51 |
| | | *Average* | **7.94** | 6.73 | 5.78 |

Table S9: Mean accuracy (%) ± standard deviation over three runs of ImageNet-pretrained Swin-T/S when transferred to 12 downstream datasets. The second row under each method indicates the number of trainable parameters (excluding the linear classifier). **T-CT improves over linear probing but underperforms LoRA.**

(a) Swin-T

| Dataset | Frozen (0) | LoRA (74832) | T-CT (532) |
|---|---|---|---|
| Arabic Characters | 83.48 ± 0.15 | **93.24 ± 0.13** | 85.02 ± 0.30 |
| Arabic Digits | 98.14 ± 0.07 | **99.19 ± 0.01** | 98.47 ± 0.04 |
| Beans | 88.28 ± 1.10 | **94.01 ± 0.37** | 89.06 ± 1.10 |
| CUB-200 | 73.42 ± 0.17 | **78.73 ± 0.28** | 74.33 ± 0.14 |
| DTD | 70.66 ± 0.13 | 70.99 ± 0.61 | **71.45 ± 0.31** |
| FashionMNIST | 89.89 ± 0.04 | **93.15 ± 0.13** | 90.23 ± 0.03 |
| FGVC-Aircraft | 48.06 ± 0.32 | **48.29 ± 0.46** | 47.58 ± 0.99 |
| Flowers102 | 86.66 ± 0.17 | **90.22 ± 0.34** | 85.35 ± 0.20 |
| Food101 | 77.05 ± 0.03 | **83.69 ± 0.11** | 78.90 ± 0.11 |
| DermaMNIST | 75.83 ± 0.27 | **76.71 ± 0.43** | 75.86 ± 0.11 |
| OCTMNIST | 69.97 ± 0.62 | **76.30 ± 1.66** | 67.97 ± 1.01 |
| PathMNIST | 89.14 ± 0.23 | **92.26 ± 0.12** | 91.73 ± 0.19 |
| *Average* | 77.69 | **82.23** | 78.53 |

(b) Swin-S

| Dataset | Frozen (0) | LoRA (148560) | T-CT (868) |
|---|---|---|---|
| Arabic Characters | 83.83 ± 0.05 | **94.38 ± 0.34** | 86.65 ± 0.50 |
| Arabic Digits | 98.28 ± 0.03 | **99.19 ± 0.05** | 98.39 ± 0.04 |
| Beans | 90.89 ± 1.95 | **95.05 ± 1.47** | 91.41 ± 0.64 |
| CUB-200 | 72.66 ± 0.56 | **79.45 ± 0.52** | 73.40 ± 0.10 |
| DTD | 69.77 ± 0.44 | 71.56 ± 0.66 | **72.43 ± 0.13** |
| FashionMNIST | 89.75 ± 0.03 | **93.52 ± 0.05** | 89.85 ± 0.08 |
| FGVC-Aircraft | 44.36 ± 0.21 | **51.94 ± 0.60** | 45.72 ± 0.27 |
| Flowers102 | 83.24 ± 0.05 | **87.67 ± 3.41** | 85.08 ± 0.25 |
| Food101 | 77.59 ± 0.06 | **85.17 ± 0.23** | 79.45 ± 0.14 |
| DermaMNIST | 76.64 ± 0.22 | **78.15 ± 0.67** | 77.14 ± 0.02 |
| OCTMNIST | 66.90 ± 0.29 | **76.97 ± 0.45** | 69.07 ± 0.60 |
| PathMNIST | 89.74 ± 0.38 | **92.79 ± 0.33** | 92.13 ± 0.15 |
| *Average* | 78.06 | **82.90** | 79.24 |

## C  Theoretical intuition

This section provides theoretical intuition behind Curvature Tuning. Section C.1 casts CT as a projection over a space of smooth functions, while Section C.2 provides a toy example illustrating how CT can improve approximation of a target function of non-vanishing curvature, upon an ideal baseline ReLU network.

### C.1  CT operates as a projection

At its core, Curvature Tuning operates by modulating the nonlinearity of the activation functions of a trained model, providing a novel approach to model steering. In order to formalize the effect of CT, the following briefly introduces the notion of spaces of smooth functions.

**Sobolev spaces**  Let $f : \mathbb{R}^d \to \mathbb{R}$ be a function and $\Omega \subseteq \mathbb{R}^d$ be a bounded domain. For $1 \leq p < \infty$, define $L^p(\Omega)$ as the space of functions $f : \Omega \to \mathbb{R}$ such that the $L^p$ norm is finite, i.e.

$$\|f\|_{L^p(\Omega)} := \left( \int_\Omega |f(\mathbf{x})|^p d\mathbf{x} \right)^{\frac{1}{p}} < \infty \tag{10}$$

Let $\alpha = (\alpha_1, \ldots, \alpha_d)$ denote a multi-index, with $|\alpha| := \sum_i^d \alpha_i$, and $\alpha_i \in \mathbb{N}, \forall i = 1, \ldots, d$. Let $q \in \mathbb{N}^*$. For $|\alpha| > 0$, define the Sobolev semi-norm

$$|f|_{W^{q,p}(\Omega)} := \left( \sum_{|\alpha| \leq q} \|D^\alpha f\|_{L^p(\Omega)}^p \right)^{\frac{1}{p}} \tag{11}$$

with $D^\alpha f := \frac{\partial^{|\alpha|} f}{\partial x_1^{\alpha_1} \ldots \partial x_d^{\alpha_d}}$ denoting $|\alpha|$-th order partial derivatives of $f$. Define the Sobolev norm

$$\|f\|_{W^{q,p}(\Omega)} := \left( \|f\|_{L^p(\Omega)}^p + |f|_{W^{q,p}(\Omega)}^p \right)^{\frac{1}{p}} \tag{12}$$

and the Sobolev space $W^{q,p}(\Omega) := \{ f : \Omega \to \mathbb{R} \text{ s.t. } \|f\|_{L^p(\Omega)}^p + |f|_{W^{q,p}(\Omega)}^p < \infty \}$.

For a finite set $\mathcal{D} = \{\mathbf{x}_i\}_{i=1}^n$, the Sobolev semi-norm becomes

$$|f|_{W^{q,p}(\mathcal{D})} := \left( \sum_{|\alpha| \leq q} \frac{1}{n} \sum_{i=1}^n \|D^\alpha f(\mathbf{x}_i)\|_p^p \right)^{\frac{1}{p}} \tag{13}$$

Finally, for $\mathbf{x} \in \mathbb{R}^d$, let $\|\mathbf{x}\|_p$ denote the $p$-norm, corresponding to the Euclidean norm for $p = 2$.

**Curvature Tuning acts as a Sobolev Projection**  To characterize Curvature Tuning, we are interested in the space $W^{2,2}(\Omega)$, equipped with the Sobolev semi-norm

$$|f|_{W^{2,2}(\Omega)}^2 = \|\nabla_{\mathbf{x}} f\|_{L_2(\Omega)}^2 + \|\nabla_{\mathbf{x}}^2 f\|_{L_2(\Omega)}^2 \tag{14}$$

We begin by considering the Sobolev semi-norm of a ReLU network (equivalent to the case of Eq. (8) with $\beta \to 1$). For each $\mathbf{x} \in \mathbb{R}^d$, the gradient of a ReLU network

$$f(\mathbf{x}) = \left( W^L \circ \varphi \circ \ldots \circ \varphi \circ W^1 \right)(\mathbf{x}) \tag{15}$$

with $[\varphi(a)]_i := \max(0, a_i)$ for $a \in \mathbb{R}^m$ and $i \in [1, m]$, is given by

$$\nabla_{\mathbf{x}} f(\mathbf{x}) = W^L \prod_{\ell=L-1}^1 D^\ell(\mathbf{x}) W^\ell \tag{16}$$

where $D^\ell(\mathbf{x})$ is a diagonal matrix with $D_{ii}^\ell(\mathbf{x}) = \mathbf{1}_{\{\mathbf{z}_i^\ell > 0\}}$, with $\mathbf{z}_i^\ell = W_i^\ell \varphi(\mathbf{z}^{\ell-1}) + \mathbf{b}_i^\ell$ denoting the pre-activation of the $\ell$-th layer, for $\ell = 1, \ldots, L$, with $\mathbf{z}^0 := \mathbf{x}$.

We make the following observations:

O1 Since ReLU networks are differentiable a. e., the gradients $\nabla_{\mathbf{x}} f(\mathbf{x})$ are bounded in norm by the network's Lipschitz constant, which can be defined as $C = \sup_{\mathbf{x} \in \Omega} \|\nabla_{\mathbf{x}} f(\mathbf{x})\|_2$. Hence, for $\Omega = \mathcal{D}$, the Lipschitz constant provides an upper bound on the first-order term of the Sobolev semi-norm in Equation 14.

O2 Finally, we observe that since ReLU networks express piece-wise affine functions, the Hessian norm vanishes a.e. (i.e. wherever the Hessian is well defined), providing a bound on the second-order term of Equation 14.

Equipped with the above observations, in the following we characterize CT by formally restating and proving Theorem 3.1.

**Theorem C.1.** *Let $f : \mathbb{R}^d \to \mathbb{R}$ denote a ReLU network, with model parameter $\mathbf{W}$ collecting all weights and biases. For $c \in [0,1]$ and fixed $\beta \in [0,1)$, replacing every instance of ReLU with a CTU (Equation 8) with hyperparameters $\beta, c$ is equivalent to projecting $f$ to a smooth function $f_{\beta,c} \in W^{2,2}(\Omega)$ in the Sobolev space $W^{2,2}(\Omega)$, with bounded Sobolev semi-norm.*

*Particularly, it holds $\|\nabla_{\mathbf{x}}^2 f(\mathbf{x})\|_{L^2(\Omega)} \leq \|\nabla_{\mathbf{x}}^2 f_{\beta,c}(\mathbf{x})\|_{L^2(\Omega)}$, from which $f_{\beta,c}$ enjoys higher local expressivity (non-vanishing curvature), while retaining the same model parameter $\mathbf{W}$.*

Before proving Theorem C.1, we state the following Lemma, bounding the derivative of a CTU.

**Lemma C.2.** *Let $\varphi_{\beta,c}(x)$ be defined according to Eq. (8), for $\beta \in [0,1)$ and $c \in [0,1]$. Then*

$$\varphi'_{\beta,c}(x) = c \left( \sigma(bx) + bx\sigma(bx)(1 - \sigma(bx)) \right) + (1-c)\sigma\left( \frac{bx}{\beta} \right) \tag{17}$$

*where $b := \frac{\beta}{1-\beta}$ and $\sigma(x) = \frac{\exp x}{1+\exp x}$ is the sigmoid activation.*

*Furthermore, $\exists\, \overline{h}_b \in \mathbb{R}^+$ such that*

$$-c\overline{h}_b \leq \varphi'_{\beta,c}(x) \leq 1 + c\overline{h}_b \qquad \forall x \in \mathbb{R}, \quad \beta \in [0,1) \tag{18}$$

*Proof.* We recall that, since $\forall x \in \mathbb{R}$, $\varphi_{\beta,c}(x)$ is defined as the convex combination of the SiLU activation function ($c = 1$) and the Softplus activation ($c = 0$), we can bound $\varphi'_{\beta,c}(x)$ by the convex combination of individual bounds obtained for the cases $c = 0$ and $c = 1$.

**Softplus**. If $c = 0$, then $\varphi'_{\beta,0}(x) = \sigma\left( \frac{x}{1-\beta} \right)$ and $0 \leq \varphi'_{\beta,0}(x) \leq 1\, \forall x$, since the derivative is defined as a sigmoid.

**SiLU**. If $c = 1$, $\varphi'_{\beta,1}(x) = \sigma(bx) + bx\sigma(bx)(1 - \sigma(bx))$. The first term in the sum is bounded by definition of sigmoid. For the second term, we note that $\sigma(bx)(1 - \sigma(bx))$ is also bounded, and achieves it maximum at $x = 0$, for which $0 \leq \sigma(bx)(1 - \sigma(bx)) \leq \frac{1}{4}$. Furthermore, in the limit $x \to +\infty$, it holds $\varphi'_{\beta,1}(x) \to 1$, while $\varphi'_{\beta,1}(x) \to 0$ for $x \to -\infty$.

In the non-asymptotic regime, $\sigma(bx)(1 - \sigma(bx)) > 0$, and so the maximum value of $bx\sigma(bx)(1 - \sigma(bx))$ also depends on $bx$. To bound $\varphi'_{\beta,c}$ in this case, let us first consider $x > 0$. By defining $\overline{h}_b = \max_{bx \geq 0} bx\sigma(bx)(1 - \sigma(bx))$, then we finally obtain $0 \leq \varphi'_{\beta,1}(x) \leq 1 + \overline{h}_b$.

For the case $x < 0$, by using the identity $\sigma(x) = 1 - \sigma(-x)$, we have that $-\overline{h}_b \leq \varphi'_{\beta,1}(x) \leq 1$. By combining the results, we have

$$-\overline{h}_b \leq \varphi'_{\beta,1}(x) \leq 1 + \overline{h}_b \qquad \forall x \in \mathbb{R}, \quad \beta \in [0,1) \tag{19}$$

In conclusion, by convex combination of cases $c = 0$ and $c = 1$, Eq. (19) holds uniformly in $x$. $\square$

We can now prove Theorem C.1. To do so, for $f_{\beta,c}$ we have to show that

1. $f_{\beta,c}$ is smooth in $\mathbf{x}$, for $\mathbf{x} \in \Omega$
2. $\|f_{\beta,c}\|_{W^{2,2}(\Omega)} < \infty$

for a network $f_{\beta,c}$ obtained by replacing every ReLU $\varphi$ with a CTU $\varphi_{\beta,c}$, while keeping all learned parameters $\mathbf{W}$ fixed.

*Proof.* We provide a proof for $\Omega = \mathcal{D} = \{\mathbf{x}_i\}_{i=1}^n$, under the common i.i.d. assumption on $\mathcal{D}$.

To prove the first point, we observe that for $\beta \in [0, 1)$, the CTU activation function is smooth, i.e. $\varphi_{\beta,c} \in \mathcal{C}^\infty(\mathbb{R})$, thus making the whole network $f_{\beta,c}$ smooth.

We now consider the Sobolev semi-norm $|f_{\beta,c}|_{W^{2,2}(\Omega)}$. Starting with the first-order gradient, by recalling that CT replaces each occurrence of ReLU with the CTU activation function (Equation 8), the input gradient of CT is given by

$$\nabla_{\mathbf{x}} f_{\beta,c}(\mathbf{x}) = W^L \prod_{\ell=L-1}^{1} D_{\beta,c}^\ell(\mathbf{z}^\ell) W^\ell \tag{20}$$

where $D_{\beta,c}^\ell(\mathbf{z}^\ell) = \mathrm{diag}(\varphi_{\beta,c}'(\mathbf{z}^\ell))$ with $\varphi_{\beta,c}'(\mathbf{z}^\ell)_i := \varphi_{\beta,c}'(\mathbf{z}_i^\ell)$ according to Eq. (17).

To bound the Jacobian norm, we observe that

$$\|\nabla_{\mathbf{x}} f_{\beta,c}(\mathbf{x})\| = \|W^L \prod_{\ell=L-1}^{1} D_{\beta,c}^\ell(\mathbf{z}^\ell) W^\ell\| \tag{21}$$

$$\leq \|W^L\| \prod_{\ell=L-1}^{1} \|D_{\beta,c}^\ell(\mathbf{z}^\ell)\| \|W^\ell\| \tag{22}$$

$$\leq \|W^L\| \prod_{\ell=L-1}^{1} \sqrt{d_\ell}(1 + c\overline{h}_b)\|W^\ell\| < \infty \qquad \text{(Lemma C.2)} \tag{23}$$

independent of $\mathbf{x}$, for $W^\ell \in \mathbb{R}^{d_\ell \times d_{\ell-1}}$, with $d_0 := d$.

We now bound the second order term. By recalling that, for every $\mathbf{x} \in \mathbb{R}^d$, the Hessian $\mathbf{H}(\mathbf{x}) = \nabla_{\mathbf{x}}^2 f_{\beta,c}(\mathbf{x})$ is symmetric positive-definite, then for $\Omega = \mathcal{D}$ it holds

$$\|\nabla_{\mathbf{x}}^2 f_{\beta,c}\|_{L_2(\mathcal{D})}^2 = \frac{1}{n} \sum_{i=1}^{n} \|\mathbf{H}(\mathbf{x}_i)\|_2^2 \leq \max_{1 \leq i \leq n} \lambda_{\max}^2(\mathbf{H}(\mathbf{x}_i)) d_\ell < \infty \tag{24}$$

with $\lambda_{\max}(\mathbf{H}(\mathbf{x}_i))$ denoting the largest singular value of $\mathbf{H}(\mathbf{x}_i)$.

Importantly, since a ReLU network $f$ has vanishing curvature a.e., then for $0 \leq \beta < 1$, we have

$$\|\nabla_{\mathbf{x}}^2 f(\mathbf{x})\| \leq \|\nabla_{\mathbf{x}}^2 f_{\beta,c}(\mathbf{x})\|.$$

Lastly, we note that, whenever $\Omega$ is a finite discrete set $\mathcal{D}$, $f_{\beta,c}$ is measurable, ensuring that $\|f_{\beta,c}\|_{W^{2,2}(\Omega)} < \infty$, concluding the proof. □

Theorem C.1 shows that CT operates by projecting a ReLU network $f$ to a smooth function $f_{\beta,c}$ in a restricted Sobolev space. Crucially, $f_{\beta,c}$ enjoys bounded gradients (and so is well behaved), and non-vanishing local-curvature for $0 < \beta < 1$, making it locally more expressive than the affine spline $f$, for fixed $\mathbf{W}$.

Furthermore, for fixed $(\beta, c)$, CT indeed operates as a projection, since replacing every ReLU with $\varphi_{\beta,c}$ is idempotent. Importantly, while for the original ReLU network $f \in W^{2,2}(\Omega)$ the derivatives $D^\alpha f$ are understood in a weak-sense, for $c \in [0, 1]$ and $\beta \in [0, 1)$, $f_{\beta,c}$ belongs to a Sobolev space $W_{\mathrm{str}}^{2,2}(\Omega) \subset W^{2,2}(\Omega)$ of smooth functions, whereby the derivative $D^\alpha f_{\beta,c}$ are understood in the strong (i.e. classical) sense.

We leave for future work extending our result to T-CT, which is associated with a non-convex optimization problem of finding optimal $(\beta, c)$ for every neuron in the network. An additional important direction is to more closely compare $\|\nabla_{\mathbf{x}} f\|$ and $\|\nabla_{\mathbf{x}} f_{\beta,c}\|$, which may reveal more precise Lipschitz behaviour for CT, potentially better guiding the search for $\beta$ and $c$.

## C.2 Toy example

We conclude the discussion by providing the full derivation for the motivating example in Section 3.

Consider a binary classification problem in $\mathbb{R}^2$, whereby one is given two classes $\{\mathbf{x} \in \mathbb{R}^2 : \|\mathbf{x}\|_2 \leq \frac{1}{2}\}$ and $\{\mathbf{x} \in \mathbb{R}^2 : \frac{3}{2} \leq \|\mathbf{x}\|_2 \leq 2\}$. The decision boundary maximizing the margin between the two classes is given by $S^1 = \{\mathbf{x} \in \mathbb{R}^2 : \|\mathbf{x}\| = 1\}$.

For a ReLU network $f : \mathbb{R}^2 \to \mathbb{R}$, the maximum margin boundary is recovered by assigning $f(\mathbf{x}) = 0 \; \forall \mathbf{x} \in S^1$, for which $\sigma(f(\mathbf{x})) = 0.5$. To measure the approximation error $e$, the boundary is parameterized by $\boldsymbol{\gamma}(t) = (\cos 2\pi t, \sin 2\pi t)$, for $t \in [0, 1]$.

Then, the error is expressed by the line integral $e = \int_{\gamma} |f| d\mathbf{x} = \int_0^1 |f(\boldsymbol{\gamma}(t))| \|\boldsymbol{\gamma}'(t)\| dt$. Since $f$ is an Affine Spline Operator, and each linear region in $\Omega$ is convex, then the integral along $\gamma$ can be broken down into the integral along the intersection of $\gamma$ with the spline partition $\Omega$, i.e. $\Omega_\gamma := \Omega \cap S^1$. Importantly, this allows us to pull back the affine spline breakpoints from $\Omega_\gamma$ to $[0, 1]$, so that $0 \leq t_1 \leq \ldots \leq t_{r'} \leq 1$, where $r' = |\Omega_\gamma|$. And we augment the breakpoints with the end points so that $0 = t_0 \leq t_1 \leq \ldots \leq t_{r'} \leq t_{r'+1} = 1$. Then,

$$e = \int_0^1 |f(\boldsymbol{\gamma}(t))| \|\boldsymbol{\gamma}'(t)\| dt \tag{25}$$

$$= 2\pi \sum_{k=0}^{r'} \int_{t_k}^{t_{k+1}} |\mathbf{A}_{r_k,\cdot} \boldsymbol{\gamma}(t) + \mathbf{b}_{r_k}| dt \tag{26}$$

$$= 2\pi \sum_{k=0}^{r'} \int_{t_k}^{t_{k+1}} (-1)^{z_k(t)} \left( \mathbf{A}_{r_k,\cdot} \boldsymbol{\gamma}(t) + \mathbf{b}_{r_k} \right) dt \tag{27}$$

with $z_k(t) := \mathbf{1}_{\{\mathbf{A}_{r_k,\cdot} \boldsymbol{\gamma}(t) + \mathbf{b}_{r_k} < 0\}}$, where $r_k$ denotes which spline region the $k$-th segment $[t_k, t_{k+1}]$ falls into. Then,

$$e = 2\pi \sum_{k=0}^{r'} \int_{t_k}^{t_{k+1}} (-1)^{z_k(t)} \left( \mathbf{A}_{r_k,1} \cos 2\pi t + \mathbf{A}_{r_k,2} \sin 2\pi t + \mathbf{b}_{r_k} \right) dt \tag{28}$$

$$= 2\pi \sum_{k=0}^{r'} \left( \int_{t_k}^{s_k} (-1)^{z_k(t)} g'_{r_k}(t) dt + \int_{s_k}^{t_{k+1}} (-1)^{z_k(t)} g'_{r_k}(t) dt \right) \tag{29}$$

$$= 2\pi \sum_{k=0}^{r'} \left( (-1)^{z_k(t_k)} \left[ g_{r_k}(t) \right]_{t_k}^{s_k} + (-1)^{z_k(t_{k+1})} \left[ g_{r_k}(t) \right]_{s_k}^{t_{k+1}} \right) \tag{30}$$

where

$$g'_{r_k}(t) = \mathbf{A}_{r_k,1} \cos 2\pi t + \mathbf{A}_{r_k,2} \sin 2\pi t + \mathbf{b}_{r_k},$$

$$g_{r_k}(t) = \mathbf{A}_{r_k,1} \frac{\sin 2\pi t}{2\pi} - \mathbf{A}_{r_k,2} \frac{\cos 2\pi t}{2\pi} + \mathbf{b}_{r_k} t,$$

and $s_k \in [t_k, t_{k+1}]$ is defined so $z_k(t)$ holds the same value for $t \in [t_k, s_k]$ and the opposite for $t \in (s_k, t_{k+1}]$. If for $t \in [t_k, t_{k+1}]$, $z_k(t)$ holds the same value, then simply set $s_k = t_k$.

Then since both $(-1)^{z_k(t_k)} \left[ g_{r_k}(t) \right]_{t_k}^{s_k}$ and $(-1)^{z_k(t_{k+1})} \left[ g_{r_k}(t) \right]_{s_k}^{t_{k+1}}$ are non-negative, it is clear $t_{k+1} \to t_k \quad \forall k \implies e \to 0$.

Hence, assuming the ReLU network considered attained optimal approximation error $e > 0$, reducing the error further requires increasing the number of breakpoints of the ASO, in turn requiring a degree of retraining (either through PEFT or training from scratch). With this view, Curvature Tuning opens an additional avenue for model adaptation: steering the model's decision boundaries by modulating the nonlinearity of the activation function, allowing to tune a model towards optimality without expensive retraining. To this end, it is important to note that modulating decision boundaries is orthogonal to feature adaptation and finetuning, since it allows to change the shape of decision boundaries while keeping the model parameter $\mathbf{W}$ fixed.

## D  Curvature Tuning (CT) implementation

The following code provides the Python implementation for S-CT and T-CT:

- SCTU & TCTU: classes that define the CTU module used in S-CT and T-CT, respectively.
- `replace_module` & `replace_module_dynamic`: functions that apply the appropriate module replacement to integrate S-CT or T-CT into a model.

```python
import torch
from torch import nn
import torch.nn.functional as F

class SCTU(nn.Module):
    """
    CTU for Steering CT.
    """
    def __init__(self, shared_raw_beta, shared_raw_coeff, threshold
    =20):
        super().__init__()
        self.threshold = threshold
        self._raw_beta = shared_raw_beta
        self._raw_coeff = shared_raw_coeff
        self._raw_beta.requires_grad = False
        self._raw_coeff.requires_grad = False

    @property
    def beta(self):
        return torch.sigmoid(self._raw_beta)

    @property
    def coeff(self):
        return torch.sigmoid(self._raw_coeff)

    def forward(self, x):
        beta = torch.sigmoid(self._raw_beta)
        coeff = torch.sigmoid(self._raw_coeff)
        one_minus_beta = 1 - beta + 1e-6
        x_scaled = x / one_minus_beta

        return (coeff * torch.sigmoid(beta * x_scaled) * x +
                (1 - coeff) * F.softplus(x_scaled, threshold=self.
                threshold) * one_minus_beta)
```

```python
class TCTU(nn.Module):
    """
    CTU for Trainable CT.
    """
    def __init__(self, num_input_dims, out_channels, raw_beta=1.386,
    raw_coeff=0.0, threshold=20):
        super().__init__()
        self.threshold = threshold

        # Decide channel dim based on input shape
        if num_input_dims == 2 or num_input_dims == 3:  # (B, C) or (B
        , L, D)
            channel_dim = -1
        elif num_input_dims == 4: # (B, C, H, W)
            channel_dim = 1
        else:
            raise NotImplementedError(f"Unsupported input dimension {
            num_input_dims}")
```

```python
        param_shape = [1] * num_input_dims
        param_shape[channel_dim] = out_channels

        # Init beta
        self._raw_beta = nn.Parameter(torch.full(param_shape, float(
        raw_beta)))

        # Init coeff
        self._raw_coeff = nn.Parameter(torch.full(param_shape, float(
        raw_coeff)))

    @property
    def beta(self):
        return torch.sigmoid(self._raw_beta)

    @property
    def coeff(self):
        return torch.sigmoid(self._raw_coeff)

    def forward(self, x):
        beta = torch.sigmoid(self._raw_beta)
        coeff = torch.sigmoid(self._raw_coeff)
        one_minus_beta = 1 - beta + 1e-63
        x_scaled = x / one_minus_beta

        return (coeff * torch.sigmoid(beta * x_scaled) * x +
                (1 - coeff) * F.softplus(x_scaled, threshold=self.
                threshold) * one_minus_beta)
```

```python
def replace_module(model, old_module=nn.ReLU, new_module=SCTU, **
kwargs):
    """
    Replace all instances of old_module in the model with new_module.
    """
    device = next(model.parameters(), torch.tensor([])).device  #
    Handle models with no parameters

    # Replace modules
    for name, module in model.named_modules():
        if isinstance(module, old_module):
            ct = new_module(**kwargs).to(device)

            # Replace module in the model
            names = name.split(".")
            parent = model
            for n in names[:-1]:
                if n.isdigit():
                    parent = parent[int(n)]  # for Sequential/
                    ModuleList
                else:
                    parent = getattr(parent, n)

            last_name = names[-1]
            if last_name.isdigit():
                parent[int(last_name)] = ct  # for Sequential/
                ModuleList
            else:
                setattr(parent, last_name, ct)

    return model
```

```python
def replace_module_dynamic(model, input_shape, old_module=nn.ReLU,
new_module=TCTU, **kwargs):
    """
```

```python
    Replace all instances of old_module in the model with new_module
    that's dynamically created based on the number of output channels.
    """
    device = next(model.parameters(), torch.tensor([])).device
    dummy_input = torch.randn(*input_shape).to(device)

    module_metadata = {}  # name -> (num_input_dims, out_channels)
    hooks = []

    def make_hook(name):
        def hook(module, input, output):
            num_input_dims = input[0].dim()
            if num_input_dims in (2, 3):      # (B, C) or (B, L, D)
                out_channels = output.shape[-1]
            elif num_input_dims == 4:          # (B, C, H, W)
                out_channels = output.shape[1]
            else:
                raise NotImplementedError(f"Unsupported output shape {
                output.shape} in {name}")
            module_metadata[name] = (num_input_dims, out_channels)

        return hook

    # Register hooks to all modules of the target type
    for name, module in model.named_modules():
        if isinstance(module, old_module):
            hooks.append(module.register_forward_hook(make_hook(name))
            )

    # Run dummy forward pass
    model(dummy_input)

    # Clean up hooks
    for hook in hooks:
        hook.remove()

    # Replace modules
    for name, module in model.named_modules():
        if isinstance(module, old_module) and name in module_metadata:
            num_input_dims, out_channels = module_metadata[name]
            ct = new_module(num_input_dims=num_input_dims,
            out_channels=out_channels, **kwargs).to(device)

            # Replace module in the model
            names = name.split(".")
            parent = model
            for n in names[:-1]:
                if n.isdigit():
                    parent = parent[int(n)]  # for Sequential/
                    ModuleList
                else:
                    parent = getattr(parent, n)

            last_name = names[-1]
            if last_name.isdigit():
                parent[int(last_name)] = ct  # for Sequential/
                ModuleList
            else:
                setattr(parent, last_name, ct)

    return model
```

# E    LoRA implementation

The following code provides the Python implementation of LoRA used in Section 4:

- `LoRALinear` & `LoRAConv2d`: classes that define LoRA-enhanced versions of the `Linear` and `Conv2d` modules.
- `get_lora_model`: a function that replaces all `Linear` and `Conv2d` modules in a model with their corresponding LoRA versions.

```python
import torch
from torch import nn as nn
from torch.nn import functional as F

class LoRALinear(nn.Module):
    """
    A Linear layer that applies LoRA to a frozen, pretrained Linear.
    """

    def __init__(self, original_layer: nn.Linear, r: int = 4, alpha:
    float = 1.0):
        super().__init__()
        self.in_features = original_layer.in_features
        self.out_features = original_layer.out_features
        self.r = r
        self.alpha = alpha

        # Freeze the original layer's parameters
        self.weight = nn.Parameter(original_layer.weight.data,
        requires_grad=False)
        if original_layer.bias is not None:
            self.bias = nn.Parameter(original_layer.bias.data,
            requires_grad=False)
        else:
            self.bias = None

        # LoRA parameters B and A
        # B: [out_features, r]
        # A: [r, in_features]
        self.B = nn.Parameter(torch.zeros((self.out_features, r)))
        self.A = nn.Parameter(torch.zeros((r, self.in_features)))

        # Initialize LoRA weights
        nn.init.kaiming_uniform_(self.B, a=5 ** 0.5)
        nn.init.zeros_(self.A)

    def forward(self, x):
        # Normal forward with the frozen weight
        result = F.linear(x, self.weight, self.bias)

        # LoRA path: B @ A
        # shape of BA = [out_features, in_features]
        # Then F.linear with BA
        lora_update = F.linear(x, (self.alpha / self.r) * (self.B @
        self.A))

        return result + lora_update

class LoRAConv2d(nn.Module):
    """
    A Conv2d layer that applies LoRA to a frozen, pretrained Conv2d.
    """
```

```python
    def __init__(self, original_layer: nn.Conv2d, r: int = 4, alpha:
float = 1.0):
        super().__init__()

        self.out_channels = original_layer.out_channels
        self.in_channels = original_layer.in_channels
        self.kernel_size = original_layer.kernel_size
        self.stride = original_layer.stride
        self.padding = original_layer.padding
        self.dilation = original_layer.dilation
        self.groups = original_layer.groups
        self.bias_available = (original_layer.bias is not None)

        self.r = r
        self.alpha = alpha

        # Freeze original parameters
        self.weight = nn.Parameter(original_layer.weight.data,
        requires_grad=False)
        if self.bias_available:
            self.bias = nn.Parameter(original_layer.bias.data,
            requires_grad=False)
        else:
            self.bias = None

        # Flattened shape for weight is [out_channels, in_channels *
        k_h * k_w]
        k_h, k_w = self.kernel_size
        fan_in = self.in_channels * k_h * k_w  # Flattened input dim

        # Define LoRA parameters: B and A
        # B: [out_channels, r]
        # A: [r, fan_in]
        self.B = nn.Parameter(torch.zeros((self.out_channels, r)))
        self.A = nn.Parameter(torch.zeros((r, fan_in)))

        # Initialize LoRA weights
        nn.init.kaiming_uniform_(self.B, a=5 ** 0.5)
        nn.init.zeros_(self.A)

    def forward(self, x):
        # Standard (frozen) convolution
        result = F.conv2d(
            x,
            self.weight,
            bias=self.bias,
            stride=self.stride,
            padding=self.padding,
            dilation=self.dilation,
            groups=self.groups
        )

        # Compute LoRA update
        # 1) Flatten conv kernel in the same manner as above
        # 2) Multiply B and A -> shape [out_channels, in_channels *
        k_h * k_w]
        # 3) Reshape it back to [out_channels, in_channels, k_h, k_w]
        BA = self.B @ self.A  # shape [out_channels, fan_in]

        # Reshape to conv kernel
        k_h, k_w = self.kernel_size
        lora_weight = BA.view(
            self.out_channels,
            self.in_channels,
            k_h,
```

```python
            k_w
        ) * (self.alpha / self.r)

        # Perform conv2d with the LoRA weight (no extra bias term for
        LoRA)
        lora_update = F.conv2d(
            x,
            lora_weight,
            bias=None,
            stride=self.stride,
            padding=self.padding,
            dilation=self.dilation,
            groups=self.groups
        )

        return result + lora_update
```

```python
def get_lora_model(model: nn.Module, r: int = 4, alpha: float = 1.0):
    """
    Recursively replace all Conv2d and Linear modules in model with
    LoRA-enabled versions. Freezes original weights and adds LoRA
    parameters.
    """
    for name, child in list(model.named_children()):
        # If child is a Conv2d, replace it with LoRAConv2d
        if isinstance(child, nn.Conv2d):
            lora_module = LoRAConv2d(child, r=r, alpha=alpha)
            setattr(model, name, lora_module)

        # If child is a Linear, replace it with LoRALinear
        elif isinstance(child, nn.Linear):
            lora_module = LoRALinear(child, r=r, alpha=alpha)
            setattr(model, name, lora_module)

        else:
            # Recursively traverse children
            get_lora_model(child, r=r, alpha=alpha)

    return model
```

