# OpenReview forum: "Curvature Tuning: Provable Training-free Model Steering From a Single Parameter"
_NeurIPS.cc/2025/Conference — NeurIPS 2025 poster_

### Official Review · Reviewer_Njg7 · 2025-06-26

**Clarity:** 4
**Significance:** 2
**Originality:** 3
**Rating:** 4
**Confidence:** 3

**Summary:**

This paper introduces a method for adapting pre-trained neural networks to
downstream tasks through the modification of activation functions. The main
advantage of their method, in comparison to existing popular techniques such as
LoRA, is its parameter efficiency, introducing at most two additional parameters
per activation layer. The method is motivated by its ability to preserve
non-zero curvature when applied to neural network blocks that conform to the
'max-affine spline function' class. The method is evaluated on two distinct
architectures: ResNets and Swin Vision Transformers. The authors compared two
variants of their proposed method (CT and Train CT) against LoRA. Train CT
exhibited superior performance in the ResNet experiments, whereas LoRA was more
effective when applied to Swin Vision Transformers.

**Questions:**

- The paper emphasizes interpretability in the abstract and introduction as a
  key advantage of the CT method. However, it was not clear to me in the
  experiments how interpretability was demonstrated. Could you give a specific
  example of how the claimed interpretability is helpful when fine-tuning on
  realistic problems?
- Can CT be used e.g. for GRPO? Can CT be used for language model fine-tuning?
  It would be great to see some experiments in this context.
- You mentioned that you had to pretrain Swin from scratch because it used GELU
  and not ReLU. You also mention that SiLU is compatible with CTU, so can you
  perform some experiments fine-tuning on a SiLU model? (E.g. Qwen3 and Llama
  3.2 use SiLU)
- You replace ReLUs with CTUs with $c=0.5$. Why this value in particular? Is this
  value in some sense the closest to ReLU? It is not clear to me despite looking
  at Figure 3.
- Can you add results to Table 1 for LoRA, where you have tuned $r$ and
  $\alpha$? This would allow us to frame the performance of CT correctly.

**Ethical Concerns:**

["NO or VERY MINOR ethics concerns only"]

**Final Justification:**

I would like to increase my score, as I believe I was previously not taking into account the relevance of the LoRA r=1 baseline presented by the authors. I think this new framing presents their method as a viable and cheap alternative to LoRA, thus deserving of acceptance.

There are still some minor improvements to make. I would still like to see more ablations/baselines (e.g. tune the activation function with very obvious parameters) to further justify the presented theory as strictly necessary from a practical point of view. I would also like to see some experiments combining LoRA with Trainable CT.

**Limitations:**

Authors are missing a few limitations. Around their experiments, in particular
they are missing experiments with other modalities (e.g. text), and larger scale
experiments. Their method appears limited in that it is only better than (untuned) LoRA on
the ResNet.

**Quality:**

2

**Strengths And Weaknesses:**

Strengths:

- The paper is well written, and the CT method is elegantly explained. I enjoyed reading Sections 2 and 3 in particular!
- The method is very relevant to the community, as the ability to efficiently
  fine-tune models is a key requirement for many applications.
- Both proposed methods are interesting, with CT method potentially allowing
  incredibly cheap fine-tuning, and Train CT being orders of magnitude more
  parameter efficient vs. existing methods.

Weaknesses:

- Theorem 3.1 might benefit from extra clarity on the exact meaning of
  'non-vanishing curvature'. Is it lower-bounded by something sensible? If so,
  how does the bound behave as we vary $\beta$/$c$/model size?
- Interpretability is highlighted extensively in the abstract/introduction, but
  in what exact sense, nor how this is useful in practice is not really discussed.
- Experiments are only conducted on image tasks, e.g. there are no language
  benchmarks.
- Results are disappointing: while the method shows improvement vs. LoRA on the ResNet architecture (Table
  1), it does not show an improvement on the Swin ViT (Table 3).
- Invalid baselines: the benefit of Train CT over LoRA is deceptive, since LoRA
  is only rank 1, and $\alpha$ has not been tuned.
- Authors mention that CT performs poorly with Swin, potentially because the
  pretraining dataset they used, Imagenette, is small. This hypothesis can be
  tested by e.g. pretraining Swin on a larger dataset (ImageNet perhaps). Or
  alternatively by finding an pretrained image ViT that uses SiLU or ReLU.

Overall, while the Methods were well presented, I unfortunately found the experimental results unconvincing, and that they hinder the quality of paper.

Other smaller points:

- The paper introduces aconyms without defining them (e.g. DNs (line 90), MASO (line 141), VQ (line 126))
- typo line 253: "achieves better performance then LoRA"
- missing error bars in Table 1 and 3.

---

> ### Author Rebuttal · Authors · 2025-07-31
>
> We thank the reviewer for the constructive feedback!  Before addressing specific concerns, we briefly restate the central aim of our paper: to establish CT as a highly efficient approach to parameter tuning, which is orthogonal and complementary to established parameter-centric PEFT methods. In our experimental evaluation, we achieve results that are comparable to other PEFT methods, but with considerably less parameters.
> ## Local Curvature
> We thank the reviewer for suggesting an interesting theoretical comparison: understanding the impact on the approximation error of increasing model size vs tuning curvature of a fixed-size model. In the following, we discuss the implications of Theorem 3.1 for fixed-size models, and leave further theoretical analysis to future work.
> The proposed CT method is grounded in the theory of Affine Spline Operators (ASOs). By tuning the curvature of each activation function, CT allows to increase local expressivity of a model (i.e. local curvature).
> Importantly, while the hypothesis space of ASOs parameterized by sufficiently deep ReLU networks enjoys universal approximation ability [1], Th 3.1 shows that once a model size is fixed, affine spline operators are characterized by zero (i.e. vanishing) local curvature a.e. in their domain.
> Importantly, to approximate functions of non-zero local curvature, increasing model size of ASOs provably yields non-zero error in the non-asymptotic limit, due to their local affine nature. This is further discussed in the motivating example in Section 3, as well as Appendix C.2.
> ## Interpretability of CT
> As acknowledged by reviewers cxvD and W2ME, CT is interpretable in that it exposes two learnable parameters per neuron, which provably modulate the activation function’s curvature. In turn, this controls the curvature of decision boundaries, which is intimately tied to both generalization and robustness [2,3].
> To see this, we note that for a deep network $f$ , the decision boundary of any class $i$ and $j$ is given by the set of $
> \{ x \in R^d : g(x) := f_i(x) − f_j (x) = 0 \} $, for $i \ne j$. Particularly, $g$ is itself a deep network, sharing the same parameters as $f$ up until the penultimate layer. Hence, modulating non-linearity of activation functions via $\beta$ directly controls the curvature of both model function and its decision boundaries, which is provably connected to generalization and robustness [2,3]. We are happy to add a paragraph to the main paper clarifying this relationship and its connection to interpretable curvature tuning.
> ## Performance on Transformers
> The paper posits two potential reasons for the underperformance of Trainable CT vs LoRA on ReLU Swin-T/S models pretrained on Imagenette:
> 1. Trainable CT uses significantly fewer parameters than LoRA (0.71% and 0.58% on Swin-T/S).
> 2. The small size of the Imagenette pretraining dataset results in weaker representations.
>
> To investigate the second factor, we conducted an additional experiment using ImageNet-pretrained Swin-T/S models, with GELU activations (equivalent to CTU with $\beta=0.6403$ and $c=1$).
>
> |Dataset|Swin-T|||Swin-S|||
> |------------------|--------|------|---------------|--------|------|---------------|
> |                 |Frozen|LoRA|Trainable CT|Frozen|LoRA|Trainable CT|
> |Arabic Characters|77.62|**93.57**|86.10|75.30|**94.58**|85.06|
> |Arabic Digits|98.06|**99.12**|98.04|98.10|**99.18**|97.88|
> |Beans|88.28|**95.31**|92.19|87.50|**92.97**|89.84|
> |CUB-200|73.59|**76.82**|72.30|72.16|**77.68**|73.04|
> |DTD|**71.17**|69.73|70.85|70.16|70.48|**72.82**|
> |FashionMNIST|89.75|**92.95**|89.10|89.81|**93.45**|86.68|
> |FGVC-Aircraft|39.48|47.16|**47.73**|34.47|**52.42**|45.09|
> |Flowers102|84.52|83.25|**85.41**|81.36|81.43|**85.04**|
> |Food101|77.05|**82.30**|78.02|77.72|**84.47**|75.89|
> |DermaMNIST|76.51|**77.11**|75.21|76.66|**77.41**|75.46|
> |OCTMNIST|67.00|**76.70**|67.30|66.80|**77.00**|64.30|
> |PathMNIST|90.65|**92.56**|91.24|89.89|**91.91**|90.33|
> |**Average**|77.81|**82.22**|79.46|76.66|**82.75**|78.45|
>
> **Results:** *Trainable CT* achieves an average relative improvement over the frozen baseline of 2.92% (Swin-T) and 3.76% (Swin-S). The gap with LoRA is considerably narrower (3.19% and 5.42%) than that observed in the ReLU experiment on Imagenette. This supports our second hypothesis: the earlier performance gap was partly due to weaker representations from small-scale pretraining, which amplified the advantage of LoRA’s higher parameter count.
> These new results suggest that *Trainable* CT can achieve competitive performance while using only 0.71% and 0.58% of LoRA’s trainable parameters, underscoring its strong parameter efficiency. Importantly, CT and LoRA operate from orthogonal perspectives and can be combined for improved performance—our goal is not to outperform LoRA directly, but to offer a complementary PEFT method. Additionally, this experiment showcases the generality of Train CT: which covers GELU-based models.
>
> ## CT vs. Higher-rank and Tuned-$\alpha$ LoRA
> We extend our LoRA comparison to higher ranks and tuned $\alpha$ values on ImageNet-pretrained ResNet-18 and Imagenette-pretrained Swin-T (ReLU-based). More specifically, we test LoRA with $r \in {1, 2, 4}$ and $\alpha \in {r, 2r, 4r}$ and report the best performance.
>
> | Dataset | ResNet-18 |  |  |  | Swin-T |  |  |  |
> |---|---|---|---|---|---|---|---|---|
> |  | Trainable CT | LoRA $r=1$ | LoRA $r=2$ | LoRA $r=4$ | Trainable CT | LoRA $r=1$ | LoRA $r=2$ | LoRA $r=4$ |
> | Arabic Characters | 93.76 | 93.37 | 95.06 | **96.19** | 41.95 | 56.32 | 68.36 | **77.86** |
> | Arabic Digits | 99.03 | 99.08 | **99.22** | **99.22** | 90.82 | 97.54 | 98.28 | **98.62** |
> | Beans | 94.01 | 93.23 | 96.09 | **97.66** | 68.49 | 75.52 | 80.47 | **82.81** |
> | CUB-200 | **64.30** | 54.83 | 56.45 | 58.15 | 6.09 | 7.42 | 9.03 | **11.36** |
> | DTD | **63.62** | 54.36 | 56.17 | 57.77 | 17.04 | 16.99 | 20.64 | **20.74** |
> | FashionMNIST | 91.07 | 91.65 | 92.79 | **93.42** | 77.07 | 83.90 | 86.83 | **88.73** |
> | FGVC-Aircraft | **46.44** | 29.19 | 32.43 | 38.76 | 6.14 | 5.59 | 5.91 | **7.41** |
> | Flowers102 | **86.55** | 67.53 | 68.92 | 72.89 | 16.53 | 16.66 | **20.36** | 19.91 |
> | Food101 | 66.04 | 64.40 | 66.00 | **66.75** | 15.20 | 18.17 | 21.54 | **25.80** |
> | DermaMNIST | **77.66** | 74.21 | 76.46 | 77.36 | 71.37 | 74.08 | 75.41 | **76.16** |
> | OCTMNIST | 69.53 | 74.27 | **77.30** | 76.10 | 53.23 | 63.53 | 66.80 | **71.40** |
> | PathMNIST | 87.17 | 87.62 | **89.36** | 89.28 | 77.35 | **81.31** | 79.97 | 76.16 |
> | **Average** | **78.26** | 73.64 | 75.52 | 76.96 | 45.11 | 49.75 | 52.80 | **54.75** |
>
> **Results:** On ResNet-18, *Train CT* outperforms LoRA with rank 1/2/4 by 10.20%, 6.63%, and 3.22%, respectively, indicating that CT remains highly effective even when compared to stronger LoRA baselines. While for Swin-T, *Trainable CT* trails LoRA (rank 1/2/4) by 8.29%, 16.30% and 21.17%, respectively.
>
> ## Setting $c=0.5$ for CTUs
> The rationale for choosing $c = 0.5$ is explained in the last paragraph of Sec 3.1. Since the SiLU and SoftPlus tend to shift the mean of the activations in opposite directions (negative for SiLU and positive for SoftPlus), setting $c = 0.5$ balances these effects and helps mitigate the shift, making the mean output closer to ReLU.
>
> To demonstrate why this is beneficial, we conducted an additional ablation comparing Steering CT vs SiLU- and SoftPlus-only variants (both reparameterized by CTU) on ResNet-18 and ResNet-50:
> |Dataset|ResNet-18|||ResNet-50|||
> |-------|---------|----|--------|---------|----|--------|
> ||CT|SiLU|SoftPlus|CT|SiLU|SoftPlus|
> |Arabic Characters|**87.65**|81.96|84.89|83.66|**86.46**|81.10|
> |Arabic Digits|**98.77**|97.99|98.64|98.37|**98.56**|98.41|
> |Beans|**90.36**|88.80|88.02|**91.93**|86.46|91.15|
> |CUB-200|**63.18**|62.92|63.08|64.62|65.06|**65.61**|
> |DTD|62.66|62.89|**63.24**|66.91|67.18|**67.20**|
> |FashionMNIST|88.70|88.52|**88.93**|90.34|89.92|**90.35**|
> |FGVC-Aircraft|**38.68**|36.31|37.44|**41.16**|37.91|40.09|
> |Flowers102|81.97|80.37|**83.14**|**83.84**|83.77|83.69|
> |Food101|62.27|61.34|**62.76**|68.02|**68.04**|67.90|
> |DermaMNIST|75.05|74.63|**75.11**|**75.89**|75.74|75.79|
> |OCTMNIST|**67.27**|64.90|66.57|**68.00**|67.30|67.00|
> |PathMNIST|87.51|86.81|**87.84**|**90.26**|90.03|**90.26**|
> |**Average**|**75.34**|73.95|74.97|**76.92**|76.37|76.55|
>
> **Results:** The SiLU version trails steering CT by 1.98% and 0.94% on ResNet-18 and ResNet-50, respectively. And the SoftPlus version trails steering CT by 0.53% and 0.54%, respectively. These results highlight the performance degradation caused by the output shift in simpler formulations, and the effectiveness of balancing the two.
> Additionally, we observed that the average optimal $\beta$ for CT is 0.84 (ResNet-18) and 0.94 (ResNet-50), while for SiLU and SoftPlus, the values are closer to 1—0.99/0.95 (SiLU) and 0.94/0.98 (SoftPlus)—suggesting more limited flexibility for steering. This provides further support for the necessity of setting $c = 0.5$, enabling broader adaptability.
>
> ## Applicability of CT in More Contexts
> At its core, CT provides a mechanism for steering model behavior and is not tied to any specific modality. The steering CT can be applied directly in any domain via a lightweight grid search procedure, while *Trainable CT* only requires a task-specific objective, similar to other PEFT methods such as LoRA. Given this flexibility, we believe CT can be readily extended to domains beyond vision. While we focused on image classification in this paper, we agree that applying CT to language model finetuning (e.g., for GRPO) is an exciting direction for future work.
>
>
> ## References
> [1] Universal Function Approximation by Deep Neural Nets with Bounded Width and ReLU Activations. Hanin. Mathematics. 2019.
>
> [2] Spectrally-normalized margin bounds for neural networks. Bartlett, et al. NeurIPS 2017.
>
> [3] Robustness via Curvature Regularization, and Vice Versa. Moosavi-Dezfooli, et al; CVPR, 2019

---

> > ### Comment · Area_Chair_n95j · 2025-08-05
> >
> > Dear Reviewer Njg7,
> >
> > The authors have updated their detailed responses to your concerns. As the Author-Reviewer discussion period comes to an end, please confirm whether your concerns were addressed and whether this may inform any changes to your score.
> >
> > Best regards,
> >
> > AC

---

> ### Comment · Reviewer_Njg7 · 2025-08-05
>
> Thank you for the detailed response to my review.
>
> >We are happy to add a paragraph to the main paper clarifying this relationship and its connection to interpretable curvature tuning.
>
> Please, I think this would aid the manuscript.
>
> I am not convinced by the results under the heading 'Performance on Transformers'. Quite often the performance seems to degrade versus the frozen model.
>
> ---
>
> In my original review, I said:
>
> >Invalid baselines: the benefit of Train CT over LoRA is deceptive, since LoRA is only rank 1, and
>  has not been tuned.
>
> While it is good to see more thorough LoRA results, I'm still not convinced by the results. It would be simpler to take some existing results, or cite a thorough methodology, e.g. [1]
>
> [1]: LoRA+: Efficient Low Rank Adaptation of Large Models
>
> ---
>
> >CT and LoRA operate from orthogonal perspectives and can be combined for improved performance
>
> Are you able to show that the methods can be combined, and that an improvement is obtained?
>
> ---
>
> Overall, I appreciate the clarifications given by the authors regarding my questions. Unfortunately I am still unconvinced by the results: (a) the results are only on image problems, (b) the method isn't applicable with general activation function, (c) there is no comparison/ablation with a 'dumb' baseline (e.g. we add the most obvious learned parameter that we can, e.g. relu(\beta x)), (d) Trainable CT reasonably often degrades performance.

---

> ### Author Response · Authors · 2025-08-08
>
> We thank the reviewer for engaging in the discussion, and for providing detailed feedback. Before replying to individual points, we would like to reiterate the core design philosophy and advantages of Curvature Tuning.
>
> 1. The proposed method strives for extreme parameter efficiency, whereby up to two parameters are injected per activation neuron. As such, rather than focusing on beating SOTA on finetuning, our experiments center around evaluating downstream generalization and robustness in settings with constrained parameter counts, whereby we strive to provide **the most parameter efficient method**.
> 2. Curvature Tuning operates orthogonally to parameter-centric adaptation methods, in that it modulates the curvature of activation functions by changing the functional form of their nonlinearity. In line with our parameter efficiency goal, CT takes advantage of the functional form of ReLU, and proposes a flexible reparameterization whose hypothesis space encompasses several popular activations, including GeLU, SiLU, and Softplus, making it broadly applicable to modern finetuning settings.
> 3. Thanks to the theoretical connection with Affine Spline Operators (Sec 3), the method provably modulates curvature, providing considerable benefits on downstream robustness benchmarks.
> By modulating activation functions, the paper opens a novel avenue for parameter-efficient finetuning.
>
> In the following, we reply to each raised point.
>
> ## Target Activation Functions
> While it is in principle possible to design activation-agnostic methods, e.g. via Taylor expansions, this approach would violate one of the core design principles of Curvature Tuning, namely to inject as few parameters as possible into the model.
>
> By taking advantage of the functional form of ReLU and its smoothed versions, we propose a novel reparameterization that encompasses several popular activation functions, including ReLU, SiLU/Swish, GeLU and Softplus.
>
> In fact, what on the one hand the method may lose in terms of generality, is on the other hand gained in terms (1) explicit curvature modulation and (2) parameter efficiency.
>
> In stark contrast, parameter-centric adaptation methods cannot operate on activation functions and the associated non-linearity, and entail non-interpretable design choices, such as adapter placement and rank.
>
> ## Choice of Baselines
> We note that the proposed naive baseline, ReLU($\beta x$), falls in the realm of parameter-centric adapters, which as discussed above are unable to change the functional form of the non-linearity.
> An alternative, simple, naive baseline is expressible by CTUs. As discussed in Sec 3, for $c=1$ and $\beta = 0$, one obtains $x \mapsto \frac{1}{2} x$ which is a linear function with zero curvature globally. Such a baseline is naive in that it offers maximum curvature regularization but limited expressivity, and is orthogonal to parameter-centric methods.
> ## Importance of $r=1$ LoRA Baseline
> We would like to briefly justify and clarify the choice of low rank for the LoRA adapters. Since Curvature Tuning operates orthogonally (and thus complementarily) to parameter-centric finetuning methods, our experimental evaluation was designed to showcase the method’s efficiency in budget constrained settings. Where CT injects up to two parameters per activation neuron, LoRA with $r=1$ and no biases uses at least twice the parameter count, making it a fair baseline w.r.t. efficiency.
>
> ## Performance on GeLU Transformers
> We thank the reviewer for pointing out that *Trainable CT* sometimes underperforms the frozen model. After careful inspection, we found that the learning rate inherited from the ReLU-based Swin-T/S setup in our original submission was too high for the GeLU configuration.
> We provide below an updated evaluation on GELU-based Swin-T/S experiments, whereby we perform cross-validation to select the optimal learning rate for each method (i.e., the baseline, Trainable CT, and LoRA). We report the best results for each method below:
> |Dataset|Swin-T|||Swin-S|||
> |-|-|-|-|-|-|-|
> ||Frozen|LoRA|Trainable CT|Frozen|LoRA|Trainable CT|
> |Arabic Characters|83.27|**93.57**|86.10|83.78|**94.58**|86.76|
> |Arabic Digits|98.24|**99.12**|98.39|98.32|**99.18**|98.39|
> |Beans|89.84|**95.31**|92.19|**92.97**|**92.97**|92.19|
> |CUB-200|73.65|**77.60**|74.23|72.61|**79.98**|73.42|
> |DTD|71.17|70.32|**71.86**|70.16|70.48|**72.82**|
> |FashionMNIST|89.75|**92.95**|90.25|89.85|**93.45**|89.96|
> |FGVC-Aircraft|47.61|47.16|**47.73**|44.52|**52.42**|45.09|
> |Flowers102|86.88|**90.57**|85.41|83.28|**90.29**|85.04|
> |Food101|77.05|**83.23**|78.97|77.72|**85.50**|79.60|
> |DermaMNIST|76.51|**77.11**|75.76|76.66|**77.41**|**77.41**|
> |OCTMNIST|69.10|**76.70**|67.30|67.00|**77.00**|69.70|
> |PathMNIST|90.65|**92.56**|91.84|89.89|**92.60**|92.08|
> |**Average**|79.48|**83.02**|80.00|78.90|**83.82**|80.21|
>
> **Results:** *Trainable CT* outperforms the baseline on 9 out of 12 datasets for Swin-T, and on 11 out of 12 datasets for Swin-S.

---

### Official Review · Reviewer_W2ME · 2025-07-02

**Clarity:** 3
**Significance:** 4
**Originality:** 4
**Rating:** 5
**Confidence:** 3

**Summary:**

This paper introduces a novel perspective on fine-tuning by shifting the focus from conventional weight adaptation to activation function modulation—an often overlooked yet critical component in determining the model's expressivity and decision boundary.
To this end, the authors propose Curvature Tuning (CT), a method designed to finetune the curvature of the activation functions. CT theoretically ensures smoother decision boundaries and empirically demonstrates improved generalization and robustness across various downstream tasks.

**Questions:**

1.	Limited Architectural Diversity in Evaluation

To what extent does Curvature Tuning (CT) generalize across diverse neural network architectures beyond ResNet and Swin Transformers? Have the authors considered evaluating CT on a broader range of model families (e.g., VGG, MobileNet, GoogLeNet, DenseNet, or WideResNet) to substantiate its architectural robustness and flexibility?

2.	Incomplete Robustness Evaluation

While Table 1 presents a broad comparison, why are Trainable CT and LoRA omitted from the robustness experiments? Would including these baselines help more rigorously validate CT’s claimed advantage under distributional shifts?

3.	Limited Experimental Scope

Can the authors clarify whether Curvature Tuning (CT) is applicable beyond vision tasks? Have they considered evaluating its generalization performance in other domains such as NLP, multimodal learning, or reinforcement learning to demonstrate broader applicability?

4.	Additional Clarification on Class Imbalance

How does CT perform under class-imbalanced conditions? Has its robustness to skewed label distributions been empirically evaluated, and if not, would this be a worthwhile direction to explore given real-world dataset imbalances?

**Ethical Concerns:**

["NO or VERY MINOR ethics concerns only"]

**Final Justification:**

I have reviewed the submitted rebuttal, and many of my concerns have been resolved through the extensive experiments provided. I recommend acceptance of the paper.

**Limitations:**

Yes

**Quality:**

3

**Strengths And Weaknesses:**

Strengths

1.	Novel Perspective on Model Steering

The paper proposes a fundamentally new fine-tuning paradigm by modulating activation functions, as opposed to the conventional weight-centric approaches. This enables direct control over the curvature of decision boundaries and, by extension, the model's hypothesis space.

2.	Sound Theoretical Grounding

The proposed method is underpinned by a rigorous theoretical framework. Specifically, it interprets deep neural networks as compositions of max-affine spline operators (MASO) and formulates Curvature Tuning (CT) as a projection onto a smooth function space, thereby providing analytical clarity and mathematical justification.

3.	Parameter Efficiency

The method introduces minimal overhead: the Steering CT variant introduces only a single hyperparameter, while the Trainable CT variant requires significantly fewer parameters than rank-1 LoRA, yet achieves superior performance.

4.	Interpretability

The impact of curvature control is both mathematically tractable and visually interpretable. The adjustment of the 棺 parameter provides an intuitive handle for regulating the non-linearity and expressive capacity of the model.

5.	Strong Empirical Performance

CT consistently outperforms established baselines such as linear probing and LoRA across 12 diverse downstream classification tasks. It also demonstrates marked gains in robustness, achieving state-of-the-art results on robustness benchmarks such as RobustBench.


Weaknesses

1.	Limited Architectural Diversity in Evaluation

While the theoretical foundation of Curvature Tuning (CT) leverages Max-Affine Spline Operators (MASOs), which generalize a broad class of activation functions including ReLU and Leaky ReLU, the empirical validation is confined to ResNet and Swin Transformer backbones. To substantiate the architectural generality of CT, further evaluation on diverse model families—such as VGG, GoogLeNet, MobileNet, DenseNet, or WideResNet—would be beneficial, even if not complete.

2.	Incomplete Robustness Evaluation

While Table 1 provides a comprehensive comparison across baseline methods, the robustness experiments lack inclusion of key variants such as Trainable CT and LoRA. This omission makes it difficult to fully assess the robustness advantage of Curvature Tuning (CT) under distributional shifts. Including these comparisons would strengthen the empirical claim of CT’s superior robustness.

---

> ### Author Rebuttal · Authors · 2025-07-31
>
> We thank the reviewer for their thorough and encouraging feedback! In response, we conducted an additional experiment and provided detailed clarifications addressing the reviewer’s concerns. Before delving into specific points, we would like to reiterate the central aim of our paper: to establish CT as a highly efficient approach to parameter tuning, which is orthogonal and complementary to established parameter-centric PEFT methods. In our experiments, CT achieves competitive performance with significantly greater parameter efficiency.
>
> ## Weakness 1: Limited Architectural Diversity in Evaluation
> To demonstrate CT's applicability to broader model families, we conducted an additional experiment applying *Trainable CT* to an ImageNet-pretrained VGG11 (with batch normalization). We compared it with linear probing and LoRA ($r=1$, $\alpha=1$):
>
> |Dataset|Frozen|LoRA|Trainable CT|
> |------------------|--------|------|---------------|
> |Arabic Characters|81.07|89.20|**93.07**|
> |Arabic Digits|98.00|98.83|**99.06**|
> |Beans|91.41|**92.97**|**92.97**|
> |CUB-200|61.44|59.30|**63.22**|
> |DTD|**64.89**|64.36|64.47|
> |FashionMNIST|89.49|**90.63**|90.33|
> |FGVC-Aircraft|39.51|40.14|**47.49**|
> |Flowers102|80.08|77.59|**84.57**|
> |Food101|60.95|64.51|**66.44**|
> |DermaMNIST|76.31|73.87|**78.50**|
> |OCTMNIST|66.10|**73.80**|70.20|
> |PathMNIST|86.67|**89.64**|88.38|
> |**Average**|74.66|76.24|**78.23**|
>
> **Results:** *Trainable CT* achieves an average relative improvement of 5.55% over linear probing and 3.44% over LoRA, demonstrating strong performance even on architectures not originally considered in the paper.
>
> ## Weakness 2: Incomplete Robustness Evaluation
> We omitted LoRA from the robustness experiments because applying it to improve robustness would introduce confounders compared to Steering CT. Indeed, methods such as LoRA require an explicit optimization target—such as minimizing cross-entropy loss for classification tasks. In contrast, Steering CT can improve robustness by simply adjusting the curvature of the model decision boundary, without relying on labeled data or explicit loss functions. Similarly, while Trainable CT directly modulates the curvature of decision boundaries (and thus bears a direct effect on adversarial robustness [1]), a direct and fair comparison with LoRA would require incorporating adversarial training techniques, such as generating adversarial examples using PGD, which would somewhat obfuscate the isolated effect of Trainable CT.
>
> Beyond the above experiment, deriving methods specifically tailored around CT that enable more targeted defenses against distributional shifts or adversarial threats is an exciting direction for future work.
>
> ## Question 1: Limited Architectural Diversity in Evaluation
> See Weakness 1.
>
> ## Question 2: Incomplete Robustness Evaluation
> See Weakness 2.
>
> ## Question 3: Limited Experimental Scope
> CT is not limited to vision tasks. The steering CT can be applied directly in any domain via a lightweight grid search procedure, while *Trainable CT* only requires a task-specific objective, similar to other PEFT methods such as LoRA. Therefore, we anticipate that CT is applicable to other domains, including NLP, multimodal learning, and reinforcement learning. Exploring these directions would be an intriguing avenue for future research.
>
> ## Question 4: Additional Clarification on Class Imbalance
> We do include class-imbalanced datasets in our experiments—most notably DermaMNIST, where the most frequent class accounts for 67% and the least for only 1% of training samples. As shown in Table 1 and Appendix Table 4 of our paper, *Trainable CT* consistently outperforms all considered baselines on this dataset across ResNet-18/50/152. This suggests that CT maintains robustness under skewed label distributions.
>
> We sincerely appreciate the reviewer’s thoughtful and constructive comments. We hope that our responses and the additional experiment have adequately addressed the raised concerns and further highlighted the strengths and generality of our method.
>
> ## References
> [1] Robustness via Curvature Regularization, and Vice Versa. Moosavi-Dezfooli, et al; CVPR 2019.

---

> > ### Comment · Area_Chair_n95j · 2025-08-05
> >
> > Dear Reviewer W2ME,
> >
> > The authors have updated their detailed responses to your concerns. As the Author-Reviewer discussion period comes to an end, please confirm whether your concerns were addressed and whether this may inform any changes to your score.
> >
> > Best regards,
> >
> > AC

---

> ### Author Response · Authors · 2025-08-07
>
> To further showcase the direct impact of curvature tuning on adversarial robustness, we extend the RobustBench evaluation to *Trainable CT* and LoRA, each finetuned in the vanilla setting, without introducing adversarial examples. This experiment better elucidates the direct impact of each method on robustness, without introducing confounders such as training with adversarial samples or external regularization.
>
> Specifically, we transfer ImageNet-pretrained ResNet-18/50/152 models to CIFAR-10/100 using the same setup as in our main paper—linear probing (i.e., Frozen in the table), *Trainable CT*, and LoRA . We then evaluate the adversarial robustness of the resulting models under an $\ell_\infty$ attack using RobustBench. The results are as follows:
>
> | Model      | Dataset   | Frozen | LoRA | Trainable CT |
> |------------|-----------|--------|------|---------------|
> | ResNet-18  | CIFAR10   | 0.30   | 0.70 | **1.57**       |
> |   | CIFAR100  | 0.03   | 0.07 | **0.17**       |
> |   | Average   | 0.17   | 0.38 | **0.87**       |
> | ResNet-50  | CIFAR10   | 0.20   | 0.33 | **2.43**       |
> |   | CIFAR100  | 0.00   | 0.03 | **0.07**       |
> |   | Average   | 0.10   | 0.18 | **1.25**       |
> | ResNet-152  | CIFAR10   | 0.43   | 0.20 | **5.10**       |
> |  | CIFAR100  | **0.17**   | 0.00 | 0.00       |
> |  | Average   | 0.30   | 0.10 | **2.55**       |
>
> **Results:** On ResNet-18/50/152, *Trainable CT* achieves an average relative improvement of 420.00% / 1150.00% / 750.00% over linear probing. In stark contrast, LoRA provides a slight boost in robustness, while at times has detrimental effects with 130.00% / 83.33% / –66.67%, respectively. These results indicate that even without explicit adversarial training, *Trainable CT* can substantially enhance $\ell_\infty$ adversarial robustness, since it directly modulates decision boundary curvature, whereas LoRA offers limited or even negative improvements, since it leaves activation function nonlinearity unchanged. This empirical finding further supports the practical advantage of  *Trainable CT* w.r.t. robustness: by directly modulating the curvature of decision boundaries, it provides direct improvements on adversarial robustness, even without explicit adversarial training. We are happy to include the above evaluation as two additional columns in Table 2 in the main paper.

---

### Official Review · Reviewer_cxvD · 2025-07-03

**Clarity:** 3
**Significance:** 2
**Originality:** 3
**Rating:** 4
**Confidence:** 2

**Summary:**

This paper introduces Curvature Tuning (CT), a novel method for steering pretrained models by modifying their activation functions instead of their weights. Grounded in spline theory, CT uses a parameter, $\beta$, to modulate the curvature of the model's decision boundary. The paper presents a training-free "steering" version and a highly parameter-efficient "Trainable CT" for finetuning, demonstrating improved generalization and robustness

**Questions:**

* $\beta$ and c are often U-shaped. Is this an emergent property you find desirable? Have you tried regularizing the training to explicitly encourage or discourage this distribution?

* Your motivation for the CTU was to mitigate a mean shift from simpler formulations. Could you provide an ablation study in the appendix that shows the negative impact of this shift in deep networks?

* Given that Trainable CT used far fewer parameters than LoRA on transformers, do you think the performance gap could be closed by allocating a moderately higher parameter budget to CT?

**Ethical Concerns:**

["NO or VERY MINOR ethics concerns only"]

**Final Justification:**

My "Borderline Accept" score reflects a balance. The core idea of tuning activations is really cool and a valuable contribution, and the authors did a great job in their rebuttal. They successfully fixed a critical experimental flaw with the transformer results by correcting the learning rate, which makes the paper much stronger. However, a few things hold it back from a higher score. Even with the fix, the method still lags behind a standard LoRA baseline on transformers, and its theory is pretty focused on ReLU-based models. Ultimately, I think the paper's novelty and the successful resolution of key issues outweigh these remaining concerns, justifying its publication.

**Limitations:**

In the conclusion, the authors identify their work's primary focus on ReLU-based networks as a limitation, stating that while CT is compatible with activations like Softplus or SiLU, this area "warrants further study". They also note that a detailed, layer-wise analysis of the learned $\beta$ and c parameters in Trainable CT was not performed and remains an avenue for future research to understand representation and abstraction.

**Quality:**

3

**Strengths And Weaknesses:**

Strength:
*  The approach of tuning activations is a fresh perspective in model adaptation. Its foundation in spline theory provides a clear, interpretable mechanism for controlling model curvature, unlike more heuristic methods.

* Trainable CT achieves impressive results with significantly fewer parameters than competing methods. For instance, on ResNets, it outperformed LoRA while using only 11% to 59% of the trainable parameters.

* The method delivers substantial gains in both generalization and robustness on ResNet architectures. The improvements on downstream accuracy and against L infinity adversarial attacks are particularly noteworthy.

Weakness:

* The method's theoretical guarantees are tied to ReLU-based networks. Applying it to models with other common activations, like GELU, required modifying the base architecture for the experiments, which may limit its off-the-shelf utility.

* While effective on CNNs, Trainable CT underperformed LoRA in the transformer-based experiments. This raises questions about its effectiveness in more complex, attention-based architectures.

* The non-trainable steering version of CT requires a grid search to find the optimal $\beta$ for each task, adding a tuning step that finetuning methods avoid.

---

> ### Author Rebuttal · Authors · 2025-07-31
>
> We thank the reviewer for their thorough and encouraging feedback. In response, we conducted two additional experiments and provide below detailed clarifications to address the reviewer’s concerns.
>
> Before addressing the raised concerns, we begin by reiterating the central aim of our paper: to establish Curvature Tuning (CT) as a highly efficient approach to parameter tuning, which is orthogonal and complementary to established parameter-centric Parameter-Efficient Fine-Tuning (PEFT) methods. In our experimental evaluation, we achieve results that are comparable to other PEFT methods, but with considerable gains in terms of parameter-efficiency and novel applications to downstream robustness.
>
> ## Weakness 1: Limited Utility on Other Activations
> We agree that our theoretical guarantees are currently tied to ReLU-based networks. However, this does not preclude the application of CT to other activations. In fact, SiLU and SoftPlus are special cases of our Curvature Tuning Unit (CTU), and *Trainable CT* can be directly applied to models using these activations.
> Moreover, for GELU—commonly used in transformer models—it can be approximated by SiLU with a CTU parameterization of $\beta=0.6403$ and $c=1$. This allows us to apply *Trainable CT* to GELU-based models by initializing with these values.
> To address this point concretely, we conducted new experiments applying *Trainable CT* to ImageNet-pretrained Swin-T and Swin-S models using their original GELU activations (unlike our earlier experiments, which replaced GELU with ReLU). We compare against linear probing and LoRA ($r=1$, $\alpha=1$):
> |Dataset|Swin-T|||Swin-S|||
> |------------------|--------|------|---------------|--------|------|---------------|
> |                 |Frozen|LoRA|Trainable CT|Frozen|LoRA|Trainable CT|
> |Arabic Characters|77.62|**93.57**|86.10|75.30|**94.58**|85.06|
> |Arabic Digits|98.06|**99.12**|98.04|98.10|**99.18**|97.88|
> |Beans|88.28|**95.31**|92.19|87.50|**92.97**|89.84|
> |CUB-200|73.59|**76.82**|72.30|72.16|**77.68**|73.04|
> |DTD|**71.17**|69.73|70.85|70.16|70.48|**72.82**|
> |FashionMNIST|89.75|**92.95**|89.10|89.81|**93.45**|86.68|
> |FGVC-Aircraft|39.48|47.16|**47.73**|34.47|**52.42**|45.09|
> |Flowers102|84.52|83.25|**85.41**|81.36|81.43|**85.04**|
> |Food101|77.05|**82.30**|78.02|77.72|**84.47**|75.89|
> |DermaMNIST|76.51|**77.11**|75.21|76.66|**77.41**|75.46|
> |OCTMNIST|67.00|**76.70**|67.30|66.80|**77.00**|64.30|
> |PathMNIST|90.65|**92.56**|91.24|89.89|**91.91**|90.33|
> |**Average**|77.81|**82.22**|79.46|76.66|**82.75**|78.45|
>
> **Results:** *Trainable CT* improves over the frozen baseline by 2.92% (Swin-T) and 3.76% (Swin-S) on average, while trailing LoRA by only 3.19% and 5.42%, respectively—despite using only 0.71% and 0.58% of LoRA's trainable parameters.
>
> ## Weakness 2: Underperformance in Transformers
> In the original submission, we hypothesized two potential reasons for the underperformance of *Trainable CT* compared to LoRA on ReLU-based Swin-T/S models pretrained on Imagenette:
> 1. *Trainable CT* uses significantly fewer parameters than LoRA (0.71% and 0.58% on Swin-T/S).
> 2. The small size of the Imagenette pretraining dataset results in weaker representations, amplifying the benefit of high-parameter methods like LoRA.
>
> In our new experiment on ImageNet-pretrained, GELU-based Swin-T/S models as shown above, the number of trainable parameters for both *Trainable CT* and LoRA remains identical to the original experiment. Thus, the first factor still holds—CT continues to operate with a significantly smaller parameter budget.
> Importantly, the new results support the second hypothesis: when models are pretrained on a much larger dataset (ImageNet), the performance gap between *Trainable CT* and LoRA narrows. This suggests that the earlier underperformance was at least partly due to the limited representation capacity of models pretrained on the smaller Imagenette dataset.
> In summary, as demonstrated by the new experiment, while *Trainable CT* still trails LoRA by 3.19% (Swin-T) and 5.42% (Swin-S), it achieves this competitive performance with only 0.71% and 0.58% of LoRA's trainable parameters—highlighting its strong parameter efficiency and practical value.
> ## Weakness 3: Tuning Overhead of Steering CT
> We acknowledge that the non-trainable version of CT involves a grid search over $\beta$. However, we observe that the optimal $\beta$ values tend to be close to 1, suggesting that the search range can be substantially narrowed in practice (e.g., to [0.9, 1]) for efficiency.
> Moreover, popular PEFT methods such as LoRA also require hyperparameter tuning—for instance, selecting appropriate values for rank and $\alpha$—and typically involve grid searches as well. Thus, tuning overhead is not unique to our method.
>
> ## Question 1: U-shaped Distributions of $\beta$ and $c$
> As noted in Footnote 5 of the paper, the U-shaped distributions may be partially attributed to the sigmoid-based parameterization of $\beta$ and $c$ (which constrains them to [0,1]). Interestingly, the average values of these parameters are close to the optimal settings discovered in the steering version of CT, which we consider a desirable outcome.
> We have not explicitly regularized these distributions, as it's unclear whether such U-shapes are beneficial or detrimental. However, we agree this is an intriguing direction and plan to explore regularization effects in future work.
> ## Question 2: Negative Impact of the Shift
> To demonstrate the value of our current CTU formulation in mitigating the shift caused by simpler formulations, we compare the steering version of CT against the simpler formulation of just using SiLU or SoftPlus (both reparameterized as in CTU) on ResNet-18 and ResNet-50:
> |Dataset|ResNet-18|||ResNet-50|||
> |-------|---------|----|--------|---------|----|--------|
> ||CT|SiLU|SoftPlus|CT|SiLU|SoftPlus|
> |Arabic Characters|**87.65**|81.96|84.89|83.66|**86.46**|81.10|
> |Arabic Digits|**98.77**|97.99|98.64|98.37|**98.56**|98.41|
> |Beans|**90.36**|88.80|88.02|**91.93**|86.46|91.15|
> |CUB-200|**63.18**|62.92|63.08|64.62|65.06|**65.61**|
> |DTD|62.66|62.89|**63.24**|66.91|67.18|**67.20**|
> |FashionMNIST|88.70|88.52|**88.93**|90.34|89.92|**90.35**|
> |FGVC-Aircraft|**38.68**|36.31|37.44|**41.16**|37.91|40.09|
> |Flowers102|81.97|80.37|**83.14**|**83.84**|83.77|83.69|
> |Food101|62.27|61.34|**62.76**|68.02|**68.04**|67.90|
> |DermaMNIST|75.05|74.63|**75.11**|**75.89**|75.74|75.79|
> |OCTMNIST|**67.27**|64.90|66.57|**68.00**|67.30|67.00|
> |PathMNIST|87.51|86.81|**87.84**|**90.26**|90.03|**90.26**|
> |**Average**|**75.34**|73.95|74.97|**76.92**|76.37|76.55|
>
> **Results:** The SiLU-only version trails steering CT by 1.98% and 0.94% on ResNet-18 and ResNet-50, respectively. And the SoftPlus-only version trails steering CT by 0.53% and 0.54%, respectively. This performance gap illustrates the negative impact caused by using simpler formulations. We also observed that the average optimal $\beta$ for CT on ResNet-18/50 is 0.84/0.94. In contrast, for the other two formulations, the values are closer to 1—0.99/0.95 for SiLU and 0.94/0.98 for SoftPlus—indicating a smaller room for model steering. This reflects the negative impact of the shift from another perspective.
>
> ## Question 3: CT Performance with Higher Parameter Budget
> In our current setup, *Trainable CT* already uses the maximum number of parameters allowed under its formulation—specifically, by assigning one pair of parameters ($\beta$, $c$) to each output neuron of the activation functions throughout the network. Therefore, it is not possible to increase the parameter count within the current formulation.
> However, since *Trainable CT* is orthogonal to other PEFT methods, any additional parameter budget can be leveraged by combining it with methods like LoRA. This opens up a promising direction for achieving even stronger performance while maintaining modularity and efficiency.
>
> We sincerely appreciate the reviewer’s thoughtful and constructive comments. We hope that our responses and additional experiments have sufficiently addressed the concerns raised and clarified the strengths and contributions of our work.

---

> > ### Comment · Area_Chair_n95j · 2025-08-05
> >
> > Dear Reviewer cxvD,
> >
> > The authors have updated their detailed responses to your concerns. As the Author-Reviewer discussion period comes to an end, please confirm whether your concerns were addressed and whether this may inform any changes to your score.
> >
> > Best regards,
> >
> > AC

---

> > ### Comment · Reviewer_cxvD · 2025-08-09
> >
> > Thanks for the detailed rebuttal and all the new experiments. I've read through your responses to my questions and the discussions with the other reviewers. After consideration, I am retaining my score borderline accept.
> >
> > Your work on tuning activations is a great, fresh perspective, and I'm encouraged by the new results. The ablation study on the CTU design was a valuable addition, and the updated transformer experiments with the corrected learning rate successfully resolve the key performance degradation issue noted by Reviewer Njg7. My score reflects the balance between this novel concept and its current limitations. While the method is now shown to work on transformers, the results also confirm it still lags behind LoRA in that setting. This performance gap, combined with the core theoretical guarantees being tied to ReLU-based models, suggests more work is needed before the method has the same broad, off-the-shelf utility as established techniques.
> >
> > Overall, this is a solid paper with a very promising idea, and the reasons to accept outweigh the reasons to reject. It's a valuable contribution, and I look forward to seeing how this line of work develops.

---

> > > ### Author Response · Authors · 2025-08-09
> > >
> > > We thank the reviewer for their response, and for supporting our submission!

---

### Official Review · Reviewer_uDax · 2025-07-06

**Clarity:** 4
**Significance:** 4
**Originality:** 4
**Rating:** 5
**Confidence:** 3

**Summary:**

This paper introduces and studies curvature tuning (CT) as a novel paradigm to parameter-efficiently finetune pretrained models. Instead of finding direct modifiers over model weights, the authors suggest (learnable) modifications on the utilized activation functions. In controlled scenarios, the authors showcase provable adjustments to model decision boundaries to facilitate generalization capabilities, highlight complementarity with existing PEFT techniques, and showcase high parameter efficiency even when compared to rank 1 LoRA.

**Questions:**

See Weaknesses.

**Ethical Concerns:**

["NO or VERY MINOR ethics concerns only"]

**Final Justification:**

The authors provided additional experimental support to an already great paper, including VGG11, Swin & additional LoRA experiments. Consequently, I maintain my strong recommendation for acceptance of the paper.

**Limitations:**

Explicitly addressed in Discussions, offering sensible avenues for future research.

**Quality:**

4

**Strengths And Weaknesses:**

__Strengths:__
* The paper is really well written and structured.
* To the best of my knowledge, attacking parameter finetuning as a function of activation function tuning is novel, and incredibly interesting.
* Beyond the novelty, this paper offers a very principled perspective into the efficacy of curvature tuning, and highlight the complementary/orthogonality to existing finetuning methods.
* On its own, CT is also really parameter efficient (in parts significantly below even rank 1 LoRA).
* The experiments provided are sufficient to convince of the importance and potential general efficacy of curvature tuning for model finetuning.

__Weaknesses:__

My only gripe with this paper is the depth of comparative experiments testing CT on more models, activation functions, datasets and domains and comparing against finetuning techniques beyond LoRA (and even here, ideally also contrasting against higher rank variants). Other than that,, I am rather hard pressed to find any other major issues with this work - I find it a really interesting read, and potentially a highly impactful contribution to the model finetuning domain.

---

> ### Author Rebuttal · Authors · 2025-07-31
>
> We thank the reviewer for their thorough and encouraging feedback! As suggested, we have conducted three additional experiments to further demonstrate the effectiveness and generality of Curvature Tuning (CT). These experiments address broader model families, activation functions, and comparisons with additional LoRA configurations.
>
> We also take the opportunity to emphasize that the aim of the paper is to establish CT as a highly efficient approach to parameter tuning, which is orthogonal and complementary to established parameter-centric PEFT methods. In our experimental evaluation, we achieve results that are comparable to other PEFT methods, but with considerable gains in terms of parameter-efficiency.
>
> ## Experiment 1: CT on VGG11
> To evaluate the effectiveness of CT on a different convolutional architecture, we apply Trainable CT to an ImageNet-pretrained VGG11 model (with batch normalization). We compare it against linear probing and LoRA with $r=1$ and $\alpha=1$.
> |Dataset|Frozen|LoRA|Trainable CT|
> |------------------|--------|------|---------------|
> |Arabic Characters|81.07|89.20|**93.07**|
> |Arabic Digits|98.00|98.83|**99.06**|
> |Beans|91.41|**92.97**|**92.97**|
> |CUB-200|61.44|59.30|**63.22**|
> |DTD|**64.89**|64.36|64.47|
> |FashionMNIST|89.49|**90.63**|90.33|
> |FGVC-Aircraft|39.51|40.14|**47.49**|
> |Flowers102|80.08|77.59|**84.57**|
> |Food101|60.95|64.51|**66.44**|
> |DermaMNIST|76.31|73.87|**78.50**|
> |OCTMNIST|66.10|**73.80**|70.20|
> |PathMNIST|86.67|**89.64**|88.38|
> |**Average**|74.66|76.24|**78.23**|
>
> **Results:** *Trainable CT* achieves an average relative improvement of 5.55% over linear probing and 3.44% over LoRA, showing its strong performance even on models not originally considered in the paper.
>
> ## Experiment 2: CT on GELU-based Swin-T/S
> To test CT’s compatibility with other activation functions, we begin with GELU as a representative case. Since GELU can be approximated by SiLU, which is parameterized by CTU with $\beta=0.6403$ and $c=1$, we can apply *Trainable CT* to GELU-based models by starting with these values.
> More specifically, we apply *Trainable CT* to ImageNet-pretrained Swin-T and Swin-S, using the original GELU activations (unlike our earlier experiment in the paper, which replaced GELU with ReLU). And we compare it with linear probing and LoRA ($r=1$, $\alpha=1$).
> |Dataset|Swin-T|||Swin-S|||
> |------------------|--------|------|---------------|--------|------|---------------|
> |                 |Frozen|LoRA|Trainable CT|Frozen|LoRA|Trainable CT|
> |Arabic Characters|77.62|**93.57**|86.10|75.30|**94.58**|85.06|
> |Arabic Digits|98.06|**99.12**|98.04|98.10|**99.18**|97.88|
> |Beans|88.28|**95.31**|92.19|87.50|**92.97**|89.84|
> |CUB-200|73.59|**76.82**|72.30|72.16|**77.68**|73.04|
> |DTD|**71.17**|69.73|70.85|70.16|70.48|**72.82**|
> |FashionMNIST|89.75|**92.95**|89.10|89.81|**93.45**|86.68|
> |FGVC-Aircraft|39.48|47.16|**47.73**|34.47|**52.42**|45.09|
> |Flowers102|84.52|83.25|**85.41**|81.36|81.43|**85.04**|
> |Food101|77.05|**82.30**|78.02|77.72|**84.47**|75.89|
> |DermaMNIST|76.51|**77.11**|75.21|76.66|**77.41**|75.46|
> |OCTMNIST|67.00|**76.70**|67.30|66.80|**77.00**|64.30|
> |PathMNIST|90.65|**92.56**|91.24|89.89|**91.91**|90.33|
> |**Average**|77.81|**82.22**|79.46|76.66|**82.75**|78.45|
>
> **Results:** *Trainable CT* achieves an average relative improvement over the baseline of 2.92% (Swin-T) and 3.76% (Swin-S). While *Trainable CT* still trails LoRA by 3.19% and 5.42% respectively, the performance gap is notably smaller than in the original paper. This supports our hypothesis in the original paper that the larger gap in the earlier ReLU-based Swin experiments (pretrained on the smaller Imagenette) stemmed from underpowered representations, which gave LoRA’s greater parameter count a stronger advantage. On ImageNet-pretrained models now, *Trainable CT* achieves more comparable results **despite using just 0.71% and 0.58% of LoRA’s trainable parameters**.
>
> We also emphasize: CT is complementary to LoRA, not a replacement—beating LoRA is not our goal, but rather demonstrating CT's efficiency and compatibility.
>
> ## Experiment 3: CT vs. Higher-rank and Tuned-$\alpha$ LoRA
> We extend our LoRA comparison by evaluating higher ranks and tuned $\alpha$ values on ImageNet-pretrained ResNet-18 and Imagenette-pretrained Swin-T (ReLU-based). More specifically, we test LoRA with $r \in {1, 2, 4}$. For each rank, we use $\alpha \in \{r, 2r, 4r\}$ and report the best performance.
> |Dataset|ResNet-18||||Swin-T||||
> |------------------|------------|-------------|-------------|-------------|------------|-------------|-------------|-------------|
> |                 |Trainable CT|LoRA $r=1$|LoRA $r=2$|LoRA $r=4$|Trainable CT|LoRA $r=1$|LoRA $r=2$|LoRA $r=4$|
> |Arabic Characters|93.76|93.37|95.06|**96.19**|41.95|56.32|68.36|**77.86**|
> |Arabic Digits|99.03|99.08|**99.22**|**99.22**|90.82|97.54|98.28|**98.62**|
> |Beans|94.01|93.23|96.09|**97.66**|68.49|75.52|80.47|**82.81**|
> |CUB-200|**64.30**|54.83|56.45|58.15|6.09|7.42|9.03|**11.36**|
> |DTD|**63.62**|54.36|56.17|57.77|17.04|16.99|20.64|**20.74**|
> |FashionMNIST|91.07|91.65|92.79|**93.42**|77.07|83.90|86.83|**88.73**|
> |FGVC-Aircraft|**46.44**|29.19|32.43|38.76|6.14|5.59|5.91|**7.41**|
> |Flowers102|**86.55**|67.53|68.92|72.89|16.53|16.66|**20.36**|19.91|
> |Food101|66.04|64.40|66.00|**66.75**|15.20|18.17|21.54|**25.80**|
> |DermaMNIST|**77.66**|74.21|76.46|77.36|71.37|74.08|75.41|**76.16**|
> |OCTMNIST|69.53|74.27|**77.30**|76.10|53.23|63.53|66.80|**71.40**|
> |PathMNIST|87.17|87.62|**89.36**|89.28|77.35|**81.31**|79.97|76.16|
> |**Average**|**78.26**|73.64|75.52|76.96|45.11|49.75|52.80|**54.75**|
>
> **Results:** On ResNet-18, *Trainable CT* outperforms LoRA with rank 1/2/4 by 10.20%, 6.63%, and 3.22%, respectively, indicating that CT remains highly effective even when compared to stronger LoRA baselines. While for Swin-T, *Trainable CT* trails LoRA (rank 1/2/4) by 8.29%, 16.30% and 21.17%, respectively.
>
> ## Future Work: CT on Language Model
> As a promising direction for future work, we plan to evaluate *Trainable CT* on language models and benchmarks. This would allow us to examine CT's effectiveness in non-vision domains.
>
> ## Conclusion
> These new experiments collectively reinforce the generality and practicality of Curvature Tuning across several key aspects suggested by the reviewer:
>
> - **More Models**: Demonstrated in Experiment 1 (VGG11) and Experiment 2 (GELU-based Swin-T/S), showing that CT applies to broader model families.
> - **More Activation Functions**: Experiment 2 illustrates CT's applicability to GELU-based networks, beyond the original ReLU-based settings.
> - **More Finetuning Techniques**: Experiment 3 highlights CT’s competitiveness even against stronger LoRA configurations with higher rank and tuned alpha.
>
> We appreciate the reviewer’s insightful suggestion and hope these additions further validate CT as a compelling and complementary approach to parameter-efficient finetuning.

---

> > ### Comment · Reviewer_uDax · 2025-08-05
> > **Response to Rebuttal**
> >
> > I thank the authors for provided additional experimental support to an already great paper, including VGG11, Swin & additional LoRA experiments. Consequently, I maintain my strong recommendation for acceptance of the paper.

---

> ### Author Response · Authors · 2025-08-05
>
> We thank the reviewer for supporting our submission!

---

### Comment · Area_Chair_n95j · 2025-08-03
**Author-reviewer discussion period in progress**

Dear Reviewers,

Thank you for your efforts in reviewing this paper.

We are now in the author-reviewer discussion period. Given the detailed author responses, we encourage active discussion during this period. If you have not already, please read their response, acknowledge it in your review, and update your assessment as soon as possible.

If you have further questions or concerns, post them promptly so authors can respond within the discussion period.

Best regards,
AC

---

### Author Response · Authors · 2025-08-09
**Closing remarks**

We thank all reviewers for their thorough work and encouraging feedback, which improves the clarity and impact of our submission.

We are pleased that all reviewers found the paper to be well written, and that they further appreciated the novelty (uDax, cxvD, W2ME), theoretical soundness (uDax, W2ME), rigorousness (uDax, W2ME), interpretability (cxvD, W2ME), parameter efficiency (uDax, cxvD, Njg7), and broader relevance (Njg7) of the proposed methodology.

### Main contributions
Our paper proposes a novel perspective on Parameter-Efficient Fine-Tuning (PEFT), which is orthogonal and complementary to existing parameter-centric methods. By modulating the functional form of a model’s activation functions, the method provides a principled approach to controlling curvature of decision boundaries, with direct impact for both downstream generalization and robustness.

The method strives for extreme parameter efficiency, injecting as little as one parameter per layer (Steer CT) and two parameters per activation neuron (Trainable CT). At the same time, by modulating non-linearities, the method retains high expressivity, and encompasses popular activations such as ReLU, SiLU/Swish, Softplus, and GELU.

Crucially, the method achieves state-of-the-art results on robustness benchmarks, avoiding the need for a dedicated fine-tuning loss optimizing for robustness, overcoming limitations of existing parameter-centric methods, which do not control robustness directly.

### Summary of rebuttal and proposed changes
Our rebuttal and the following discussion addressed the following points raised by reviewers:

To be included in the main paper:

1. Paragraph further clarifying the interpretable role of curvature tuning (Njg7).
2. Extended LoRA benchmarks (uDax, cxvD, Njg7), complementing Table 1 in the main paper.
3. Extended robustness evaluation (W2ME), complementing Table 2.
4. Experiments on GeLU transformers (uDax, cxvD, W2ME), complementing Table 3.

To be added to the appendix:

1. Extended evaluation on more architectures (uDax, cxvD, W2ME).
2. Additional ablations on the design of CTUs (cxvD, Njg7).
3. Extensive learning rate cross-validation for GELU models (cxvD, Njg7).

We hope our rebuttal addresses the main concerns raised by reviewers, and thank everybody once again for the thorough discussion and insightful feedback.

---

### Note · Authors · 2025-08-12

We thank all reviewers for their thorough work and encouraging feedback, which improves the clarity and impact of our submission. We reiterate our closing remarks here for the AC's consideration.

We are pleased that all reviewers found the paper to be well written, and that they further appreciated the novelty (uDax, cxvD, W2ME), theoretical soundness (uDax, W2ME), rigorousness (uDax, W2ME), interpretability (cxvD, W2ME), parameter efficiency (uDax, cxvD, Njg7), and broader relevance (Njg7) of the proposed methodology.

## Main contributions
Our paper opens up a novel, theoretically grounded direction for Parameter-Efficient Fine-Tuning (PEFT), which is orthogonal and complementary to LoRA -- an increasingly important method in the ML landscape.

By modulating the functional form of a model’s activation functions, the method provides a principled approach to controlling curvature of decision boundaries, with direct impact for both downstream generalization and robustness.

The method strives for extreme parameter efficiency, injecting as little as one parameter per layer (Steer CT) and two parameters per activation neuron (Trainable CT). At the same time, by modulating non-linearities, the method retains high expressivity, and encompasses popular activations such as ReLU, SiLU/Swish, Softplus, and GELU.

Crucially, the method achieves state-of-the-art results on robustness benchmarks, avoiding the need for a dedicated fine-tuning loss optimizing for robustness, overcoming limitations of existing parameter-centric methods, which do not control robustness directly.

## Summary of rebuttal and proposed changes
Our rebuttal and the following discussion addressed the following points raised by reviewers:

### To be included in the main paper:

1. Paragraph further clarifying the interpretable role of curvature tuning (Njg7).
2. Extended LoRA benchmarks (uDax, cxvD, Njg7), complementing Table 1 in the main paper.
3. Extended robustness evaluation (W2ME), complementing Table 2.
4. Experiments on GeLU transformers (uDax, cxvD, W2ME), complementing Table 3.

### To be added to the appendix:

1. Extended evaluation on more architectures (uDax, cxvD, W2ME).
2. Additional ablations on the design of CTUs (cxvD, Njg7).
3. Extensive learning rate cross-validation for GELU models (cxvD, Njg7).

We hope our rebuttal addresses the main concerns raised by reviewers, and thank everyone once again for the thorough discussion and insightful feedback.

---

### Decision · Program_Chairs · 2025-09-17

**Decision:**

Accept (poster)

**Comment:**

This paper introduces a novel steering method that manipulates activation functions through a single-parameter injection, presenting an underexplored pathway in the PEFT domain. All reviewers recognize its significant contribution to the community, citing: (1) a refreshing idea, (2) strong theoretical grounding, (3) strong empirical performance (albeit under a limited scope), and (4) a well-organized manuscript with thoughtful insights.

From this AC’s perspective, the primary concerns centered on the evaluation scope and the method’s applicability, which was limited to ReLU (e.g., with all GeLUs replaced by ReLUs). The authors successfully addressed these points during the rebuttal, resolving them during the discussion.

Although some reviewers noted extra minor weaknesses, this AC believes the paper’s strengths clearly outweigh these concerns, which can be addressed in revision. This AC concurs with the reviewers’ positive assessments and recommends acceptance.